# Comments on foliated gauge theories and dualities in 3+1d

Po-Shen Hsin[1] and Kevin Slagle[1,2]

**1** Walter Burke Institute for Theoretical Physics,
California Institute of Technology, Pasadena, CA 91125, USA
**2** Institute for Quantum Information and Matter,
California Institute of Technology, Pasadena, CA 91125, USA

## Abstract

We investigate the properties of foliated gauge fields and construct several foliated field theories in 3+1d that describe foliated fracton orders both with and without matter, including the recent hybrid fracton models. These field theories describe Abelian or non-Abelian gauge theories coupled to foliated gauge fields, and they fall into two classes of models that we call the electric models and the magnetic models. We show that these two classes of foliated field theories enjoy a duality. We also construct a model (using foliated gauge fields and an exactly solvable lattice Hamiltonian model) for a subsystem-symmetry protected topological (SSPT) phase, which is analogous to a one-form symmetry protected topological phase, with the subsystem symmetry acting on codimension-two subregions. We construct the corresponding gauged SSPT phase as a foliated two-form gauge theory. Some instances of the gauged SSPT phase are a variant of the X-cube model with the same ground state degeneracy and the same fusion, but different particle statistics.



# 1 Introduction

The objective of a low-energy effective field theory is to describe the low-energy physics of a physical system while ignoring physics that occurs at higher energies, the details of which are viewed as unimportant. Restricting to low energies then reveals the universal spacetime structure that is coupled to the physics. A celebrated example of effective theory is a Fermi liquid (*e.g.* a metal) [1],which describes the low energy modes with momenta near the Fermi surface. Another example of effective theory is topological quantum field theory (TQFT), which describes the low energy physics of many gapped microscopic quantum systems and only depends on the topology of the spacetime manifold (possibly equipped with extra structure such as a spin structure if the system has neutral fermions [2]). The topological nature, such as the braiding and fusion of the excitations, has application in fault-tolerant quantum computation [3]. Effective field theories are useful in describing the low energy (IR) physics that typically occurs at large length scales, independent of many microscopic (UV) details. In particular, different microscopic models can have the same low energy physics, while the microscopic differences are washed out in the renormalization group (RG) flow. Only the universal features are captured by effective field theories.

Recently, a new kind physics exhibited by so-called fracton models [4, 5][1] necessitate a new kind of effective field theory description. The excitations in this class of models are classified by their sub-dimensional mobility: planons and lineons are restricted to move along 2D planes and 1D lines, respectively, while fractons are immobile. These excitations have spatially-dependant fusion rules, which can be formalized using a module [10]. For instance, two lineons that are constrained to move in two different directions could fuse into the vacuum when met at a point, or two neighboring fractons could fuse into a planon.[2] A gapped $D$-dimensional fracton model of length $L$ can have up to $O(L^{D-2})$ [11, 13, 14] robust zero-energy non-local degrees of freedom. This is a phenomenon of *UV/IR mixing*: the low energy physics depends on some microscopic details such as the total length measured in lattice spacing. This UV/IR mixing is captured in the recent generalization of the usual effective field theories [15–17][3] by certain singularities and discontinuities in the effective field variables, but the fields are more continuous than the variables in lattice models.

In this note we will study a related class of effective field theories called foliated quantum field theory (FQFT) (see *e.g.* [19]) that also exhibit UV/IR mixing: the fields can have discontinuities or delta function singularities on "stacks of leaves" in spacetime. The different foliations (e.g. $k = 1, 2, 3$ for three foliations) are described using a 1-form $e^k$ for each foliation $k$, which must satisfy $e^k \wedge de^k = 0$, and in this note we will assume $de^k = 0$ for simplicity. The fields are allowed to have certain kinds of singularities (detailed in Section 2.1), which are not allowed in ordinary effective field theories. These singularities embody the foliated spacetime structure [20, 21]. We will discuss examples of gapped and gapless foliated field theories. In many examples, the FQFT has the structure of coupling an ordinary gauge theory to a $\mathbb{Z}_N$ foliated gauge field, which can be thought of as a stack of BF type theories in one dimension lower.

While similar kinds of foliated fracton models are investigated using other field theories

---

[1]Fracton models were initially motivated by the glassy (*i.e.* slow) dynamics resulting from these mobility constraints. [6, 7] It was later discovered that the slow dynamics of the type-II models yields a more robust quantum memory. That is, the dynamics of the non-local degrees of freedom (which are used for the quantum memory) in Haah's code with generic time-dependant perturbations at finite (but low) temperature is asymptotically slower than for toric code. [8, 9]

[2]The latter process is the origin of the term "fracton" because in many examples a fracton is a *fraction* of a mobile particle. However, this is not always the case; for example, type-II [11] fracton models such as Haah's code [12] do not have any mobile particles (by definition).

[3]See also a recent field theory construction for type II fracton models in [18].

[17, 22–26] that implicitly depends on a foliation, our description using foliated field theory makes the dependence on the foliation more explicit, and it utilizes the foliation structure to constrain the effective action.

We remark that the mobility constraints of fracton models can be understood from gauging subsystem symmetries that only act on a subregion such on a plane. [11, 27] The gauge field of the subsystem symmetry can naturally be described using a foliated gauge field, where the symmetry acts on the leaves of the foliation [28]. The background foliated gauge field can also describe subsystem-symmetry protected topological (SSPT) phases [29, 30] using the effective action of the background foliated gauge field. We will give examples of such phases.

The note is organized as follows. In Section 2 we discuss the properties of $U(1)$ and $\mathbb{Z}_N$ foliated $n$-form gauge fields. In Section 3 we discuss a twisted foliated $\mathbb{Z}_N$ two-form gauge theory and its lattice model. In Section 4 and Section 5 we discuss two classes of models, which we call the electric and magnetic models, where we couple non-Abelian or Abelian gauge theory to foliated gauge fields, which encompass many examples of models in the literature *e.g.* [11, 31, 32] with excitations of restricted mobility. In Section 6 we discuss methods of coupling matter fields to foliated gauge fields. In Section 7 we discuss dualities between the electric and magnetic models.

There are several appendices. In Appendix A we provide an interpretation of foliated gauge theory as ordinary gauge theory but with a sum over defect insertions. In Appendix B we give a description of a foliated stack of scalars or fermions. In Appendix C we discuss an exactly solvable lattice model for an example of the model in Section 4.

## 1.1 Summary of examples

**Twisted foliated two-form gauge theory as gauged SSPT phase**   Many physical systems are protected by global symmetry, and there are invertible phases that are non-trivial only in the presence of global symmetry, known as symmetry protected topological (SPT) phases. In Section 3, we present examples of SPT phases protected by subsystem symmetry, known as subsystem SPT (SSPT) phases [29, 30]. For instance, the $\mathbb{Z}_N^x \times \mathbb{Z}_N^y \times \mathbb{Z}_N^z$ subsystem symmetry in 3+1d whose generators are supported on two-dimensional surfaces on $yz, xz, xy$ planes has an SSPT phase described by the effective action

$$\sum_{k,l} \frac{N p_{kl}}{4\pi} B^k B^l = \frac{N}{2\pi}(p_{12}B^1 B^2 + p_{13}B^1 B^3 + p_{23}B^2 B^3) , \tag{1.1}$$

where $B^k$ are background two-form $\mathbb{Z}_N$ foliated gauge field that has components $dx^k dx^\mu$ with $\mu = 0, 1, 2, 3$, $\mu \neq k$. The coefficients $p_{kl}$ are integers mod $N$ [33–36]. We construct a local commuting projector Hamiltonian model [Figure 2] for the SSPT phase.

Then we gauge the subsystem symmetries to obtain a foliated $\mathbb{Z}_N$ two-form gauge theory [(3.1)], where the gauge field $B^k$ is dynamical.[4] We also construct a lattice Hamiltonian for the gauged SPT phase [Figure 3]. We investigate the properties of the resulting two-form $\mathbb{Z}_N$ foliated gauge theory, and we find that certain examples of the theory reproduces the particle content of the $\mathbb{Z}_N$ X-cube model. The ground state degeneracy (GSD) of the foliated two-form gauge theory on a $T^3$ space with lengths $L_x, L_y, L_z$ along the three space directions measured

---

[4]The theory with the foliated gauge fields replaced by ordinary non-foliated gauge fields is discussed in [33–35, 37, 38], which is effectively an untwisted Abelian one-form gauge theory. The version with foliated gauge fields that we consider is much richer.

in the unit of a lattice cutoff is given by (3.23):

$$\text{GSD} = \prod_i q_i^{2r_i(L_x+L_y+L_z)-c_i} \tag{1.2}$$

$$c_i = 2\max(b_{12}^{(i)}, b_{23}^{(i)}, b_{13}^{(i)}) + \text{median}(b_{12}^{(i)}, b_{23}^{(i)}, b_{13}^{(i)})$$

$$b_{kl}^{(i)} = r_i - \log_{q_i} \gcd(q_i^{r_i}, p_{kl}) \,,$$

where $N = \prod_i q_i^{r_i}$ is the prime factorization of $N$. For $N = 2, p_{kl} = 1$, the ground state degeneracy equals $2^{2L_x+2L_y+2L_z-3}$, which equals the ground state degeneracy of the $\mathbb{Z}_2$ X-cube model [39]. The theory with $N = 2, p_{kl} = 1$ also has the same fusion module as the $\mathbb{Z}_2$ X-cube model. However, we show that these two theories are not the same by showing that their excitations are different.

**Coupling ordinary gauge theory to foliated two-form gauge field**   In Section 4, we consider a class of model called the electric model, with the action

$$S_E = \sum_k \frac{N}{2\pi} dA^k B^k + \frac{N}{2\pi}\eta_k(a)B^k + I(a,\phi)\,, \tag{1.3}$$

where $a$ is a $G$ gauge field for some finite or continuous group $G$, and $\eta_k \in H^2(G, \mathbb{Z}_N)$, $B^k$ is a foliated two-form gauge field satisfying $B^k e^k = 0$, and $A^k$ is a one-form gauge field. $I(a, \phi)$ describes the gauge theory of $a$, which can contain matter fields collectively denoted by $\phi$. The theory can be interpreted as coupling a $G$ gauge theory to the foliated $\mathbb{Z}_N$ two-form gauge field $B^k$ (which has a $\mathbb{Z}_N$ holonomy imposed by a Lagrangian multiplier $A^k$), using the one-form symmetry generated by $\oint \eta_k(a)$.

In Section 5, we study another class of model called the magnetic model, with the action

$$S_M = \frac{N}{2\pi}\sum_k dA^k B^k + \frac{Nq^k}{2\pi}bB^k + \frac{N}{2\pi}(da - \eta(a'))b + I(a',\phi')\,, \tag{1.4}$$

where $a'$ is a $G' = G/\mathbb{Z}_N$ gauge field, $a$ is a $\mathbb{Z}_N$ gauge field, and $\eta \in H^2(G', \mathbb{Z}_N)$ specifies the extension $G$ of $G'$ by $\mathbb{Z}_N$. $q_k$ is an integer. The part $I(a', \phi')$ describes a $G'$ gauge theory of $a'$, and it can contain matter fields collectively denoted by $\phi'$. The theory can be interpreted as coupling a $G$ gauge theory to the foliated $\mathbb{Z}_N$ two-form gauge field $B^k$ (which has a $\mathbb{Z}_N$ holonomy imposed by a Lagrangian multiplier $A^k$) using the one-form symmetry corresponding to the $\mathbb{Z}_N$ center of the extension $G$ (if the extension of $G'$ by $\mathbb{Z}_N$ is a central extension). The electric and magnetic models provide the effective field theory description for the models in *e.g.* [11, 31, 32].

**Duality**   In Section 7, we show that there is an exact duality between the electric and magnetic models,

$$
\begin{aligned}
\text{Electric:} \quad & \frac{N}{2\pi}dAB - \frac{N}{2\pi}B\eta(a') + S_{\text{top}}(a') + I[a',\phi] \\
\text{Magnetic:} \quad & \frac{N}{2\pi}d\widetilde{A}\widetilde{B} + \frac{N}{2\pi}\widetilde{b}\widetilde{B} + \frac{N}{2\pi}(d\widetilde{a} - \eta(\widetilde{a}'))\widetilde{b} + S_{\text{top}}(\widetilde{a}') + I[\widetilde{a}',\widetilde{\phi}]\,,
\end{aligned} \tag{1.5}
$$

where $(A, B)$, $(\widetilde{A}, \widetilde{B})$ describe $\mathbb{Z}_N$ foliated two-form gauge field on a single foliation with $A, \widetilde{A}$ being Lagrangian multipliers that constrain $B, \widetilde{B}$ to have $\mathbb{Z}_N$ holonomy. The ordinary gauge theory of $a'$ in the electric model and $\widetilde{a}, \widetilde{a}'$ in the magnetic model has gauge group $G_{\text{electric}}, G_{\text{magnetic}}$, respectively, related by $G_{\text{electric}} = G_{\text{magnetic}}/\mathbb{Z}_N$. $\phi, \widetilde{\phi}$ collectively denote the matter fields that couple to the $G_{\text{electric}}, G_{\text{magnetic}}$ gauge theories.

We remark that it is important that $(A, B)$ and $(\widetilde{A}, \widetilde{B})$ are foliated gauge fields in order for the duality to be valid. If $(A, B)$ and $(\widetilde{A}, \widetilde{B})$ were replaced by non-foliated gauge fields, then the electric model would become a $G_{\text{magnetic}}$ gauge theory, while the magnetic model would become a $G_{\text{magnetic}}/\mathbb{Z}_N = G_{\text{electric}}$ gauge theory, and since the gauge groups are different, the duality (1.5) would no longer hold in general.[5]

## 2  Abelian foliated gauge fields

**Notation and foliation**  To describe each foliation, we use foliation a closed (*i.e.* $de^k = 0$) one-form $e^k$ where $k$ labels the different foliations. For example, $e^k = dx, dy, dz$ for spacetime coordinates $(t, x, y, z)$. We use a subscript to indicates the degree of gauge fields; *i.e.* $u_p$ is a $p$-form gauge field. We sometimes omit the subscript to simplify the notation. We use $\mathcal{L}_k$ to denote a leaf of foliation $k$, which has codimension one. For instance, if $e^k = dz$, then $\mathcal{L}_k$ is a three-dimensional spacetime $(x, y, t)$ at some fixed $z = z_0$.

In the following discussion, $\delta(\mathcal{L})^\perp$ is a delta function one-form that has a singularity on codimension-one leaf $\mathcal{L}_k$; in other words, it is the Poincaré dual of leaf $\mathcal{L}_k$.[6] If we take $e^k = dz$, then for the leaf of foliation $k$ at $z = z_0$, it is $\delta(\mathcal{L}_k)^\perp = \delta(z - z_0)dz$. Denote $\mathcal{V}_k$ to be a codimension-0 manifold with boundary given by a leaf of foliation $k$, $\partial \mathcal{V}_k = \mathcal{L}_k$. Then $\delta(\mathcal{V}_k)^\perp$ is a zero-form *i.e.* a function, which has the property $d\delta(\mathcal{V}_k)^\perp = \delta(\mathcal{L}_k)^\perp$.[7] For instance, if $e^k = dz$, then $\mathcal{V}_k : z < z_0$, and $\delta(\mathcal{V}_k)^\perp = h(z - z_0)$ is a multiple of the step function. We will illustrate the properties of the foliated gauge fields using $U(1)$ and $\mathbb{Z}_N$ $n$-form foliated gauge theories. We also discuss the relation between the foliated gauge field and the rank-two symmetric tensor gauge field.

### 2.1  Bundles, fluxes and singularities

We will use the notation $B^k$ to denote two-forms that satisfy $B^k e^k = 0$ (for each $k$), and $A^k$ to denote gauge field with the gauge transformation $A^k \to A^k + d\lambda^k + \alpha^k$ where $\alpha^k e^k = 0$. Intuitively, the gauge field $A^k$ only has components in the directions parallel to each leaf of foliation $k$, while the gauge field $B^k$ has at least one component orthogonal to the leaf of foliation $k$. Related to this, $B_p^k$ can be intuitively understood as a gauge field $b'_{p-1}$ of one lower degree $(p - 1)$ multiplied by $e^k$, $B^k = b'_{p-1} e^k$, where $b'_{p-1}$ has mass dimension $p$ and thus it contains a degree-one delta function singularity. In the following we will describe the continuity property of the gauge parameters and the gauge fields in more details. In Appendix A we provide an interpretation of the singularities in the foliated field theories as summing over defect insertions in ordinary field theories. We remark that if an ordinary non-foliated gauge field couples to foliated gauge field, the new gauge field can have a different bundle that has similar singularity structure, as illustrated by an example in Section 5.1.1.[8]

---

[5]For instance, the $G_{\text{magnetic}}/\mathbb{Z}_N = G_{\text{electric}}$ ordinary gauge theory only has a subset of Wilson lines compared to $G_{\text{magnetic}}$ ordinary gauge theory.

[6]For general manifold $\Sigma$, it can be defined as $\int_\Sigma \omega = \int \omega \delta(\Sigma)^\perp$ for any $\omega$.

[7]This can be proven from the definition: for manifold $\Sigma$ that satisfies $\Sigma = \partial \Sigma'$, for any $\omega$ we have $\int_\Sigma \omega = \int \omega \delta(\Sigma)^\perp = \int_{\Sigma'} d\omega = \int d\omega \delta(\Sigma')^\perp = \int \omega d\delta(\Sigma')^\perp$. Thus $\delta(\Sigma)^\perp = d\delta(\Sigma')^\perp$.

[8]This is similar to the phenomenon that if an $SU(2)$ gauge field couples to a two-form gauge field by the $\mathbb{Z}_2$ center one-form symmetry, the gauge bundle changes to $SO(3)$ bundle, that is in general no longer an $SU(2)$ bundle.

**Foliated $n$-form gauge field $B_n^k$** The gauge field satisfies $B_n^k e^k = 0$, and it can have delta function one-form singularities $\delta(\mathcal{L}_k)^\perp$. It has gauge transformation

$$B_n^k \to B_n^k + d\lambda_{n-1}^k, \quad \lambda_{n-1}^k e^k = 0 , \tag{2.1}$$

where the condition $\lambda_{n-1}^k e^k = 0$ is replaced by $d\lambda_0^k e^k = 0$ when $n = 1$. The gauge parameter $\lambda_{n-1}^k$ has gauge transformation $\lambda_{n-1}^k \to \lambda_{n-1}^k + d\lambda_{n-2}^k$, and similarly for $\lambda_{n-2}^k$, until $\lambda_0^k \to \lambda_0^k + 2\pi f^k$ with $f^k$ locally an integer. The gauge parameter $\lambda_{n-1}^k$ can have delta function one-form singularities $\delta(\mathcal{L}_k)^\perp$, while the other gauge parameters of lower degree (but greater or equal to one) can also have discontinuities $\delta(\mathcal{V}_k)^\perp$, while the 0-form gauge parameter $f^k$ can only have discontinuities $\delta(\mathcal{V}_k)^\perp$.

The field strength $F_{n+1}^k = dB_n^k$ satisfies $F_{n+1}^k e^k = 0$. It is gauge invariant, and it can have singularities $\delta(\mathcal{L}_k)^\perp$. (Note $d\delta(\mathcal{L}_k)^\perp = 0$). The flux $\oint dB_n^k$ is quantized to be a multiple of $2\pi$ from the transition function $f^k$, and the flux can have discontinuities $\delta(\mathcal{V}_k)^\perp$.

**Foliated $n$-form gauge field $A_n^k$** The gauge field $A_n^k$ can have discontinuities $\delta(\mathcal{V}_k)^\perp$. For instance, if $e^k = dz$, then it can contain a step function $h(z-z_0)$. The gauge transformation is

$$A_n^k \to A_n^k + d\lambda_{n-1}^k + \alpha_n^k, \quad \alpha_n^k e^k = 0 , \tag{2.2}$$

where for $n = 0$ there is no gauge parameter $\alpha_0^k = 0$. The gauge parameters also have gauge transformations, such as $\lambda_{n-1}^k \to \lambda_{n-1}^k + d\lambda_{n-2}^k$, until $\lambda_0^k \to \lambda_0^k + 2\pi f^k$ for $f^k$ locally an integer, and $\alpha_n^k \to \alpha_n^k + d\alpha_{n-1}^k$ with $\alpha_{n-1}^k e^k = 0$ for $n > 1$ and $d\alpha_0^k e^k = 0$, until $\alpha_0^k \to \alpha_0^k + 2\pi s^k$ with $s^k$ locally an integer. The gauge parameters $\lambda_{i\geq 1}^k, \alpha_{i\geq 1}^k$ can have singularities $\delta(\mathcal{L}_k)^\perp$ and discontinuities $\delta(\mathcal{V}_k)^\perp$, while $\lambda_0^k, f^k, \alpha_0^k, s^k$ can only have discontinuities $\delta(\mathcal{V}_k)^\perp$. Note in the combination $d\lambda_i^k + \alpha_{i+1}^k$, the singularity $\delta(\mathcal{L}_k)^\perp$ in $d\lambda_i^k$ can be compensated by the same singularity in $\alpha_{i+1}^k$.

The field strength $F_{n+1}^k = dA_n^k$ has delta function singularities $\delta(\mathcal{L}_k)^\perp$ and discontinuities $\delta(\mathcal{V}_k)^\perp$. However, it is not invariant under the gauge transformation $\alpha_n^k$, while $e^k F_{n+1}^k$ is gauge invariant. The gauge invariant quantity $e^k F_{n+1}^k$ only has $\delta(\mathcal{V}_k)^\perp$ discontinuities, but not the singularity $\delta(\mathcal{L}_k)^\perp$, since $e^k \delta(\mathcal{L}_k)^\perp = 0$. The flux $\oint dA_n^k$ is defined on $n+1$-dimensional closed surfaces on leaf $k$ for the flux to be gauge invariant. It is quantized to be a multiple of $2\pi$ from the transition function $f^k$, which can contain discontinuities $\delta(\mathcal{V}_k)^\perp$. We note that due to the discontinuities in $A_n^k$, $F_{n+1}^k = dA_n^k$ may not be closed $dF_{n+1}^k \neq 0$ in the absence of operator insertion, but it has delta function one-form singularities $\delta(\mathcal{L}_k)^\perp$. On the other hand, the gauge invariant quantity $e^k F_{n+1}^k$ is closed, since $e^k \delta(\mathcal{L}_k)^\perp = 0$.

## 2.2 $U(1)$ foliated $n$-form gauge theory I

We begin by studying the simplest examples of foliated gauge theories. Consider a foliated $n$-form gauge field $B_n^k$ in $d$ spacetime dimension, which satisfies $B_n^k e^k = 0$, with the gauge transformation $B_n^k \to B_n^k + d\lambda_{n-1}^k$, where $\lambda_{n-1}^k e^k = 0$ for $n > 1$ and $d\lambda_0^k e^k = 0$. The action is given by the kinetic term for the foliated gauge field:

$$S = \frac{1}{2g^2}|dB_n^k|^2 = \frac{1}{2g^2}dB_n^k \star dB_n^k , \tag{2.3}$$

where $\star$ is the Hodge dual, and $k$ labels the foliation. We will focus on a single foliation, while the case with multiple foliations is given by copies of the theory with different $k$. We will promote the gauge coupling $g$ to be position-dependent, since coupling on different leaves can have different values. Let us analyze the symmetry and observables in the free $U(1)$ foliated $n$-form gauge theory.

### 2.2.1 Global symmetry

$U(1)$ **electric symmetry** The electric symmetry is a shift symmetry for $B_n^k$. If we turn on a background $(n+1)$-form $C_E^k$ (where the subscript stands for "electric" instead of the degree of the form), we need to replace $dB_n^k$ with $dB_n^k + C_E^k$. To be consistent with $B_n^k e^k = 0$, $C_E^k$ also satisfies $C_E^k e^k = 0$ and is therefore a foliated background gauge field. The background gauge transformation is

$$B_n^k \to B_n^k + \lambda_E^k, \quad C_E^k \to C_E^k + d\lambda_E^k, \quad C_E^k e^k = 0, \quad \lambda_E^k e^k = 0 . \tag{2.4}$$

$U(1)$ **magnetic symmetry** The magnetic symmetry is generated by $e^{i\theta \oint dB_n^k}$ with parameter $\theta \in \mathbb{R}/\mathbb{Z}$. If we turn on a background $(d-n-1)$-form $C_M^k$ for the magnetic symmetry (where the subscript stands for "magnetic" instead of the degree of the form), the action is modified via the coupling

$$\frac{dB_n^k}{2\pi} C_M^k . \tag{2.5}$$

The background gauge transformation is

$$C_M^k \to C_M^k + d\lambda_M^k + \alpha^k , \tag{2.6}$$

where $\alpha^k$ is a $(d-n-1)$-form gauge field that satisfies $\alpha^k e^k = 0$. The latter condition ensures that the coupling to $B_n^k$ is invariant under that gauge transformation $\alpha^k$, using $B_n^k e^k = 0$.

**Mixed anomaly** If we turn on background $C_E^k$, the coupling to $C_M^k$ is modified to be

$$\frac{dB_n^k + C_E^k}{2\pi} C_M^k , \tag{2.7}$$

which is not invariant under $C_M^k \to C_M^k + d\lambda_M^k$. Thus the electric and magnetic symmetries have a mixed anomaly. The anomaly can be compensated by inflow from a subsystem SPT phase in one dimension higher, with effective action

$$S_{\text{anom}} = \frac{1}{2\pi} \int dC_E^k C_M^k . \tag{2.8}$$

### 2.2.2 Observables

**Wilson $n$-surface** The Wilson $n$-surface, $e^{i \int_\gamma B_n^k}$, has nonzero support along the $n$-surface swiped by integral curves of $e^k$.[9] Suppose $e^k = dz$. Since $B_n^k e^k = 0$, the gauge field only has components that contain $dz$; thus the Wilson operator extends along the $z$ direction. [10] The Wilson operator transforms under the electric symmetry.

The Wilson operator can end on leaves of foliation $k$. To see this, we note for $n > 1$, an open Wilson $n$-surface is gauge invariant under $\lambda_{n-1}^k$ if the boundary lies on leaves of foliation $k$, since $\lambda_{n-1}^k e^k = 0$. For $n = 1$, we can parameterize $B_1^k = f \delta(\mathcal{L}_k)^\perp$ for some leaf $\mathcal{L}_k$ of foliation $k$ and function $f \sim f + 2\pi$; then $\int_\gamma B_1^k = f|_p$ where $p$ is the intersection points of $\mathcal{L}_k$ and $\gamma$, and $e^{i \int B_1^k} = e^{i f_p}$ is invariant under $f \to f + 2\pi$. We remark that the boundary of the Wilson operator does not break the electric symmetry since the generator of the symmetry,

---

[9]For instance, if $e^k = dx^1$, then the operator can only be supported on $x^1, x^{j_2}, x^{j_3} \cdots x^{j_n}$ directions, while it does not receive a contribution from the part of surfaces lying on leaves of foliation $k$.

[10]If we take $e^k = dt$, then for $n = 1$, the gauge field $B_n^k$ only has the time component, and the Wilson line $\int B_t dt$ describes a static heavy charge with the mobility class of a fracton.

which couples to $C_E^k$, has gauge redundancy of shift by $\alpha^k$ with $\alpha^k e^k = 0$ and thus it is restricted to lie on a leaf of foliation $k$. This implies that the generator of the electric symmetry that links with the Wilson operator cannot be unlinked by moving the generator passing through the boundary of the Wilson operator.

**'t Hooft $(d-n-2)$-surface**    The 't Hooft $(d-n-2)$-surface operator is defined by unit flux on the surrounding $(n+1)$-sphere, $\oint dB_n^k/2\pi = 1$. If we parametrize $B_n^k = \sum_i f_{n-1}^i \delta(\mathcal{L}_k^i)^\perp$ for some collection of leaves $\mathcal{L}_k^i$ of foliation $k$ indexed by $i$ and $(n-1)$-form gauge fields $f_{n-1}^i$ (for $n=1$ they are periodic scalars $f_0^i \sim f_0^i + 2\pi$), then the flux is $\oint_{S^{n+1}} dB_n^k = \sum_i \oint df_{n-1}^i$ with integral over the intersection of $S^{n+1}$ and $\mathcal{L}_k^i$, which is generally a $S^n$ intersection. The flux is the sum of the flux of $df_{n-1}^i$ over $S^n$ on the leaf around the 't Hooft operator, which lies on the leaf. For $n=1$ and $d=4$ spacetime dimension, the 't Hooft operator describes a planon on the leaf of foliation $k$, while for $n=2$ the 't Hooft operator is a monopole operator. The 't Hooft operator transforms under the magnetic symmetry.

## 2.3   $U(1)$ foliated $n$-form gauge theory II

Consider an $n$-form gauge field $A_n^k$ with the gauge transformation $A_n^k \to A_n^k + d\lambda_{n-1}^k + \alpha_n^k$, $\alpha_n^k e^k = 0$. For instance, if $e^k = dz$ and $n=1$, then the gauge transformation eats the $A_z$ component; thus we are left with only $A_x, A_y, A_t$ components. If the gauge field has vanishing local field strengths $F_{yz}, F_{xz}, F_{zt}$, then the theory is equivalent to a $U(1)$ gauge field in 2+1d $(x, y, t)$. Here, we will not assume this to be the case, and thus the gauge field can have $z$ dependence. The action is given by the kinetic term for the foliated gauge field:

$$S = \frac{1}{2g^2}|e^k dA_n^k|^2 = \frac{1}{2g^2}(e^k dA_n^k) \star (e^k dA_n^k) \,, \tag{2.9}$$

where $\star$ is the Hodge dual. Thus the kinetic term has contribution from the gauge field components parallel to the leaves. The index $k$ labels the foliation, and we will focus on a single foliation here, while multiple foliations are multiple copies with different $k$. We will promote the gauge coupling $g$ to be position-dependent, since the coupling on different leaves can have different values. For $e = dx^1$, the kinetic term is proportional to

$$\sum_{\mu, \nu \neq 1} (\partial_\mu A_{\nu_1, \cdots \nu_n} + \text{anti-symmetrization})(\partial^\mu A^{\nu_1, \cdots \nu_n} + \text{anti-symmetrization}) \,. \tag{2.10}$$

Let us analyze the symmetry and observables of free foliated $n$-form foliated gauge theory.

### 2.3.1   Global symmetry

$U(1)$ **electric symmetry**    The electric symmetry acts as a shift symmetry for gauge field $A_n^k$. If we turn on a background $(n+1)$-form gauge field $C_E^k$, this replaces $dA_n^k$ by $dA_n^k - C_E^k$. The background gauge transformation is

$$C_E^k \to C_E^k + d\lambda_n^k + d\alpha_n^k, \quad A_n^k \to A_n^k + \lambda_n^k + \alpha_n^k \,, \tag{2.11}$$

where $\lambda_n^k e^k$ can be nonzero.

$U(1)$ **magnetic symmetry**    The magnetic symmetry is generated by $e^{i\theta \oint dA_n^k}$ with $\theta \in \mathbb{R}/\mathbb{Z}$, which is only well-defined on a leaf of foliation $k$. The background $(d-n-1)$-form gauge field $C_M^k$ for magnetic symmetry couples to the theory as

$$\frac{dA_n^k}{2\pi} C_M^k \,, \tag{2.12}$$

where the invariance under $A_n^k \to A_n^k + \alpha_n^k$ implies $C_M^k$ is a foliated $(d-n-1)$-form gauge field that satisfies

$$C_M^k e^k = 0 \ . \tag{2.13}$$

The background gauge transformation is $C_M^k \to C_M^k + d\lambda_{d-n-2}^k$, where $\lambda_{d-n-2}^k e^k = 0$.

**Mixed anomaly**  By a similar discussion as in the previous example, the electric and magnetic symmetries have a mixed anomaly described by the subsystem SPT phase in one dimension higher

$$\frac{1}{2\pi} \int dC_E^k C_M^k \ . \tag{2.14}$$

### 2.3.2  Observables

**Wilson $n$-surface**  The Wilson operator $e^{i \oint_\gamma A^k}$ is gauge invariant under $A_n^k \to A_n^k + \alpha_n^k$ only when the $n$-surface $\gamma$ is on leaf of foliation $k$. Thus for $n = 1$ it describes a planon. The Wilson operator transforms under the electric symmetry.

**'t Hooft $(d-n-2)$-surface**  The 't Hooft operator with flux $\oint dA_n^k/2\pi = 1$ on the surrounding $n + 1$-sphere is gauge invariant when the $n + 1$-sphere lies entirely on a leaf of foliation $k$, *i.e.* when the 't Hooft operator is transverse to a leaf of foliation $k$. In other words, the 't Hooft operator is a $(d-n-3)$-dimensional locus on the leaf of foliation $k$ surrounded by the $(n+1)$-sphere, and extending along the remaining $k$th direction. Since $\oint dA_n^k$, which measures the magnetic charge, cannot move out of the leaf of foliation $k$, the 't Hooft operator can be an open ribbon operator with boundary on some leaves of foliation $k$.[11] For $n = d-3$, it describes a point operator on a leaf of foliation $k$ at some fixed time, and it can be moved along the $k$th direction.

### 2.3.3  Chern-Simons coupling and theta term

Let us discuss a deformation of the theory that does not change the local dynamics. We will focus on $d = 4$ spacetime dimensions and $n = 1$.

**Mixed theta term**  The ordinary theta angle $dA_1^k dA_1^k$ is not invariant under the gauge transformation $A_1^k \to A_1^k + \alpha_1^k$. On the other hand, the theory of gauge fields $A_1^k, B_1^k$ can have a mixed theta term. Consider the theory

$$\frac{\theta}{4\pi^2} dA_1^k dB_1^k \ . \tag{2.15}$$

The theta term leads to an analogue of Witten effect: the 't Hooft line of $A_1^k$ becomes a dyon with electric charge $\theta/2\pi$ of $B_1^k$, and the 't Hooft line of $B_1^k$ becomes a dyon with electric charge $\theta/2\pi$ of $A_1^k$.

**Chern-Simons term**  Another coupling is a Chern-Simons term on each leaf. A naive guess for such a term is

$$L_{\mathrm{CS}} \stackrel{?}{=} \frac{n_k}{4\pi} A_1^k dA_1^k \beta^k \ , \tag{2.16}$$

---

[11]This is similar to the Wilson $(d-n-2)$-surface of $\oint B_{d-n-2}$, which can also end on leaves of foliation $k$, as discussion in Section 2.2.2. As we will show in Section 2.4, the two foliated $U(1)$ gauge theories of fields $A_n^k$ and $B_{d-n-2}^k$ are in fact related by electric-magnetic duality.

where $\beta^k e^k = 0$ and it is closed and has unit period. For instance, if $e^k = dz$, we can take $\beta^k = dz/l_z$ if $z \sim z + l_z$ is compact. However, such term is not gauge invariant due to the discontinuity of $\beta^k dA_1^k$. To see this, we note that it can be written as

$$\int_\gamma A_1^k, \quad \gamma = \text{PD}(n_k dA_1^k \beta^k/4\pi), \tag{2.17}$$

where PD denotes the Poincaré dual. Thus it would be well-defined if and only if $\gamma$ is an integral cycle and lies on a leaf of foliation $k$. The last condition is satisfied since $dA_1^k \beta^k/4\pi e^k = 0$, and it is closed since $dF^k \beta^k = 0$ where $F^k$ is the field strength, $dF^k$ can be nonzero with $\delta(\mathcal{L}_k)^\perp$ singularity, but $\delta(\mathcal{L}_k)^\perp \beta^k = 0$. So the question is whether $\oint_\mathcal{V} n_k dA_1^k \beta^k/4\pi$ is an integer. This is in general not the case since it can be written as

$$\frac{n_k}{2} \int_{\gamma' \cap \mathcal{V}} \beta^k, \quad \gamma' = \text{PD}_\mathcal{V}(dA_1^k/2\pi), \tag{2.18}$$

where $\gamma'$ has boundary on a leaf of foliation $k$, since $dA_1^k$ is not closed due to the discontinuity. Thus for general $\beta^k$ that is not a delta function, the integral can be any real number. As a consequence, the coupling cannot be gauge invariant for $n_k \neq 0$.

On the other hand, we can consider the following well-defined Chern-Simons term

$$\frac{L_k}{4\pi} A_1^k dA_1^k \delta(\mathcal{L}_k)^\perp, \tag{2.19}$$

for some leaves $\mathcal{L}_k = \bigcup_i \mathcal{L}_k^{(i)}$. This is a Chern-Simons term at level $L_k$ on the chosen leaves.

## 2.4 Electric-magnetic duality for foliated $U(1)$ gauge theory

The observables and global symmetry of the $n$-form foliated $U(1)$ Maxwell theories in Section 2.2 and $(d-n-2)$-form foliated $U(1)$ Maxwell theory in Section 2.3 can be mapped to each other, with the electric Wilson operator mapped to the magnetic 't Hooft operator and vice versa. We will show that the two theories are in fact dual to each other, similar to the $S$-duality in ordinary $U(1)$ gauge theory [40, 41]. We will only sketch a derivation here.

For simplicity, let us take the spacetime manifold to have trivial topology, and $e^k = dx^k$. We begin with the foliated gauge theory I, with the action

$$\frac{1}{2g^2} F_B^k \star F_B^k + \frac{1}{2\pi}(F_B^k - dB_n^k)C_M^k, \tag{2.20}$$

where $F_B^k$ is a foliated $(n+1)$-form that satisfies $F_B^k e^k = 0$, and the Lagrangian multiplier $C_M^k$ has the gauge transformation $C_M^k \to C_M^k + \alpha_M^k$ with $\alpha_M^k e^k = 0$. Integrating out $C_M^k$ recovers the Maxwell term for the foliated gauge field $B_n$, with $F_B^k = dB_n^k$.

Alternatively, we can integrate out $F_B^k$, which imposes[12]

$$e^k(\frac{1}{g^2} \star F_B^k + \frac{1}{2\pi} C_M^k) = 0. \tag{2.21}$$

Then we find the following Lagrangian

$$\frac{1}{2\tilde{g}^2}|e^k C_M^k|^2 - \frac{1}{2\pi} dB^k C_M^k, \quad \tilde{g}^2 = -\frac{4\pi^2}{g^2}. \tag{2.22}$$

Then integrating out $B^k$ imposes $C_M^k = dA_{d-n-2}^k$, and we recover the Maxwell theory for foliated $(d-n-2)$-form gauge field $A_{d-n-2}^k$, with the gauge coupling

$$\tilde{g}^2 = -\frac{4\pi^2}{g^2}. \tag{2.23}$$

---

[12]This equation of motion also relates the gauge invariant field strengths $e^k dA_{d-n-2}^k$, $\frac{1}{g^2} \star dB_n^k$, where their singularity structures are related through the position-dependent gauge coupling.

## 2.5 $\mathbb{Z}_N$ foliated $n$-form gauge theory

Consider the following theory in $d$ spacetime dimensions

$$\frac{N}{2\pi} dA_n^k B_{d-n-1}^k \ , \tag{2.24}$$

where $B_{d-n-1}^k e^k = 0$. The gauge transformation are

$$A_n^k \to A_n^k + d\lambda_{n-1}^k + \alpha_n^k, \quad B_{d-n-1}^k \to B_{d-n-1}^k + d\lambda_{d-n-2}^k \ , \tag{2.25}$$

where $\alpha_n^k e^k = 0, \lambda_{d-n-2}^k e^k = 0$ (for $n = d-2$, it is replaced by $d\lambda_0^k e^k = 0$). The equations of motion for $A_n^k$ and $B_{d-n-1}^k$ implies that these gauge fields have $\mathbb{Z}_N$ holonomy. They describe $\mathbb{Z}_N$ foliated $n$-form gauge theory of $A_n^k$ or $(d-n-1)$-form gauge theory of $B_{d-n-1}^k$.

### 2.5.1 Global symmetry

$\mathbb{Z}_N$ **electric symmetry**   The electric symmetry is generated by the operator $e^{i\oint B_{d-n-1}^k}$. The background $(n+1)$-form gauge field $C_E^k$ couples to the theory by

$$\frac{N}{2\pi} dA_n^k B_{d-n-1}^k + \frac{N}{2\pi} B_{d-n-1}^k C_E^k \ . \tag{2.26}$$

The gauge transformation is

$$C_E^k \to C_E^k + d\lambda_E^k + \alpha_{n+1}^k, \quad A_n^k \to A_n^k + d\lambda_{n-1}^k - \lambda_E^k + \alpha_n^k, \quad B_{d-n-1}^k \to B_{d-n-1}^k + d\lambda_{d-n-2}^k \ , \tag{2.27}$$

where $\lambda_E^k, \alpha_{n+1}^k$ are background gauge transformations.

$\mathbb{Z}_N$ **magnetic symmetry**   The magnetic symmetry is generated by $e^{i\oint A_n^k}$. The background $(d-n)$-form gauge field $C_M^k$ satisfies $C_M^k e^k = 0$, and it couples as

$$\frac{N}{2\pi} dA_n^k B_{d-n-1}^k + \frac{N}{2\pi} A_n^k C_M^k \ . \tag{2.28}$$

The gauge transformation is

$$C_M^k \to C_M^k + d\lambda_M^k, \quad B_{d-n-1}^k \to B_{d-n-1}^k - \lambda_M^k + d\lambda_{d-n-2}^k, \quad A_n^k \to A_n^k + d\lambda_{n-1}^k + \alpha_n^k \ , \tag{2.29}$$

where $\lambda_M^k$ is a background gauge transformation that satisfies $\lambda_M^k e^k = 0$.

**Mixed anomaly**   The electric and magnetic symmetries have a mixed anomaly, described by the SPT phase in one dimension higher

$$\frac{N}{2\pi} \int C_E^k C_M^k \ . \tag{2.30}$$

### 2.5.2 Observables

The theory has operators

$$W = e^{i\oint A_n^k}, \quad U = e^{i\oint B_{d-n-1}^k} \ . \tag{2.31}$$

The operator $W$ is restricted to lie on a leaf of foliation $k$ in order to be invariant under the gauge transformation $\alpha_n^k$. For $n = 1$, $W$ describes a planon. The operator $W$ is charged under the electric symmetry.

The operator $U$ has $\mathbb{Z}_N$ braiding with operator $W$, and it is charged under the magnetic symmetry generated by $W$. $U$ can have boundaries if each connected component of the boundary lies on a leaf of the foliation $k$. The boundary of $U$ does not break the magnetic symmetry, since the generator $W$ of the magnetic symmetry is constrained on a leaf and cannot move out of the leaf.

### 2.5.3 Ground state degeneracy

Consider a single foliation with $e^1 = dx^1$ on a spatial $(d-1)$-torus, with lengths $l_i$ for $i = 1, \cdots d-1$ in the $x^i$ directions. To reduce the notation, we will drop the superscript $k$ in $A_n^k, B_{d-n-1}^k$. Let us fix the gauge such that the time components in $A_n, B_{d-n-1}$ vanish. The equation of motion from integrating out the time-component gauge fields $A_{0,i_2,\cdots i_{n-1}}$ and $B_{0,j_2,\cdots j_{d-n-2}}$ can be solved up to a gauge transformation by

$$A_n = \sum_{I=\{i_1,i_2\cdots i_n\}:1\notin I} q_I(t,x^1) \prod_I \frac{dx^i}{\ell_i}, \quad B_{d-n-1} = \sum_{J=\{j_1,j_2\cdots j_{d-n-2}\}:1\notin J} p_J(t,x^1) \frac{dx^1}{\ell_1} \prod_J \frac{dx^j}{\ell_j}. \tag{2.32}$$

$\sum_{I=\{i_1,i_2\cdots i_n\}:1\notin I}$ sums over all subsets $I$ of $2, 3, \cdots, n$. The effective action is then

$$S = \sum_{I,J}(-1)^{s(I,J)} \frac{N}{2\pi} \int dt \frac{dx^1}{\ell_1} \partial_0 q_I(x^1,t) p_J(x^1,t), \tag{2.33}$$

where $s(I,J) = 0, 1$, depending on the index sets $I, J$. There are $m_{n,d} \equiv C_n^{d-2} = \frac{(d-2)!}{(d-n-2)!n!}$ terms in the summation.

To obtain a finite ground state degeneracy, we can regularize the $x^1$ direction by a lattice with equal spacing $\Lambda^1$, and denote $L^1 = \ell_1/\Lambda^1$. This amounts to substituting $p_I(t,x^1) = \sum_{r=0,\cdots L^1-1} \ell_1 \delta(x^1 - r\Lambda^1) p_I^r(t)$.[13] Denote $q_I(t,r\Lambda^1) = q_I^r(t)$. Then the regularized effective action is

$$S = \sum_{r=0,\cdots(L^1-1)} \sum_{I,J}(-1)^{s(I,J)} \frac{N}{2\pi} \int dt \partial_0 q_I^r(t) p_J^r(t). \tag{2.34}$$

For each $x^1$ there is $m_{n,d} = \frac{(d-2)!}{(d-n-2)!n!}$ pairs of conjugating $\mathbb{Z}_N$ degrees of freedom, and thus the ground state degeneracy is

$$\text{GSD} = N^{m_{n,d}L^1} = N^{\frac{(d-2)!}{(d-n-2)!n!}L^1}. \tag{2.35}$$

**Topological nature of ground state degeneracy** Another way to understand the ground state degeneracy is using the operators $W = e^{i\oint A_n}$ and $U = e^{i\oint B_{d-n-1}}$. Consider a single leaf. The operator $W$ is supported on $T^n$ for coordinates $\{x^i : i \in I\}$. On each leaf there are $C_n^{d-2} = m_{n,d}$ such operators, labelled by $W_s$ with $s = 0, \cdots m_{n,d}$. The "ribbon" operator $U$ is supported on an interval in the $x^1$ direction, $[x_1^1, x_2^1]$, and extending along $\{x^j : j \in J\}$ directions. We can fix $x_1^1$, then the interval is labelled by the "ending-leaf" at $x_2^1$. For each ending-leaf there are also $m_{n,d}$ such operators, labelled by $U_s$ with $s = 0, \cdots m_{n,d}$. Denote the ground state without operator insertions by $|0\rangle$. Then a basis of the new ground states on each leaf can be obtained by acting on $|0\rangle$ with a product of $W$ wrapped on $n$ cycles in $T^{d-1}$:

$$|\{n_s\}\rangle = \prod_{s=1}^{m_{n,d}} W_s^{n_s}|0\rangle, \tag{2.36}$$

where $n_s = 0, \cdots, N-1$. Thus there are $N^{m_{n,d}}$ ground states for each leaf, and in total $N^{m_{n,d}L^1}$ ground states. Alternatively, we can describe the ground states using the basis

$$|\{\widetilde{n}_s\}\rangle = \prod_{s=1}^{m_{n,d}} U_s^{\widetilde{n}_s}|0\rangle, \tag{2.37}$$

---

[13]Note this is an allowed delta function singularity of the foliated gauge field $B_{d-n-1}$.

where $\tilde{n}_s = 0, \cdots, N-1$, and they span the $N^{m_{n,d}}$ ground states for the ending leaf at $x^1 = x_2^1$. Thus again we find $N^{m_{n,d}L^1}$ ground states in total.

Since the operators $W$ and $U$ are charged under the electric and magnetic symmetries, and the ground states transform as different eigenvalues of the symmetry, we conclude that the electric and magnetic symmetries are spontaneously broken.

## 2.6 $\mathbb{Z}_N \times \mathbb{Z}_N$ foliated gauge theory and three-loop braiding

Consider the $\mathbb{Z}_N \times \mathbb{Z}_{N'}$ foliated two-form gauge theory theory in $d = 4$ spacetime dimension

$$\frac{N}{2\pi} dA_1^k B_2^k + \frac{N'}{2\pi} dA_2'^k B_1'^k \,, \tag{2.38}$$

where the theory describes $\mathbb{Z}_N^2$ gauge fields $A_1^k$ and $B_1'^k$ after integrating out the Lagrangian multipliers $B_2^k$ and $A_2'^k$. It has a surface operator corresponding to the projective representation $H^2(\mathbb{Z}_N \times \mathbb{Z}_{N'}, U(1)) = \mathbb{Z}_{\gcd(N,N')}$, which can be denoted by $S = e^{i\frac{\text{lcm}(N,N')}{2\pi} \oint A_1^k B_1'^k}$ with $\text{lcm}(N,N')$ the least common multiple of $N, N'$. For simplicity, we only consider a single foliation $k$.

Let us consider the correlation function of $U(\Sigma) = e^{i\oint_\Sigma B_2^k}$, $W'(\Sigma') = e^{i\oint_{\Sigma'} A_2'^k}$ and $S(\Sigma'')$. Inserting $U(\Sigma)$ and $W'(\Sigma')$ implies

$$dA_1^k = -\frac{2\pi}{N}\delta(\Sigma)^\perp, \quad dB_1'^k = -\frac{2\pi}{N'}\delta(\Sigma')^\perp \,, \tag{2.39}$$

which can be solved as $A_1^k = -\frac{2\pi}{N}\delta(\mathcal{V})^\perp$, $B_1'^k = -\frac{2\pi}{N'}\delta(\mathcal{V}')^\perp$ with $\partial\mathcal{V} = \Sigma$, $\partial\mathcal{V}' = \Sigma'$. Then the correlation function is given by

$$\langle U(\Sigma)W'(\Sigma')S(\Sigma'')\rangle = e^{\frac{2\pi i}{\gcd(N,N')}\text{Tlk}(\Sigma,\Sigma',\Sigma'')} \,, \tag{2.40}$$

where $\text{Tlk}(\Sigma,\Sigma',\Sigma'') = \int \delta(\mathcal{V})^\perp\delta(\mathcal{V}')^\perp\delta(\Sigma'')^\perp$ with $\partial\mathcal{V} = \Sigma$, $\partial\mathcal{V}' = \Sigma'$ is the triple linking number of the three surfaces [42]. Thus the correlation function describes a three-loop braiding process [43] between the operator $S$ and the magnetic surface operators $U = e^{i\oint B_2^k}$, $W' = e^{i\oint A_2'^k}$ by a similar computation as in [44] (see also [45] for a discussion using lattice model).[14] In terms of the gauge fields on the leaves, the operators $U, V, S$ have the following interpretation (see Figure 1). The operator $U$ corresponds to a string consisting of the magnetic particle excitation in the $\mathbb{Z}_N$ one-form gauge theories on a collection of leaves. The operator $V$ corresponds to the domain wall operator in the $\mathbb{Z}_{N'}$ two-form gauge theory on the leaves. The operator $S$ intersects the leaves by the $\mathbb{Z}_N$ Wilson line operator of charge $\text{lcm}(N,N')k/N'$ for the $\mathbb{Z}_N$ one-form gauge field on the leaves, in the $k$th vacuum of the $\mathbb{Z}_{N'}$ two-form gauge theory on the leaves.

We remark that one can also consider $\mathbb{Z}_N \times \mathbb{Z}_N \times \mathbb{Z}_N$ Dijkgraaf-Witten theory

$$\frac{N}{2\pi} dA_1^k B_2^k + \frac{N}{2\pi} dA_2^k B_1^k + \frac{N}{2\pi} dA_1'^k B_2^k + \frac{Np}{(2\pi)^2} A_1^k dA_1'^k B_1^k \,, \tag{2.41}$$

where the first three terms describe $\mathbb{Z}_N^3$ gauge fields $A_1^k, B_1^k, A_1'^k$ after integrating out the Lagrangian multipliers $B_2^k, A_2^k, A_2'^k$. For simplicity, we take a single foliation $k$. By a similar computation as in $e.g.$ [47], there is three-loop braiding between the magnetic surface operators for $\mathbb{Z}_N$ gauge fields.

---

[14]One can show by a similar computation as in [44] that in any spacetime dimension (including 2+1d), decoupled untwisted $\mathbb{Z}_N$ $n$-form gauge theory and $m$-form gauge theory has a similar correlation function involving three operators: the magnetic operators for the $n, m$-form gauge theory, and the electric operator $\int a_n a'_m$ with $a_n, a'_m$ being the $n$-form and $m$-form $\mathbb{Z}_N$ gauge fields [46].

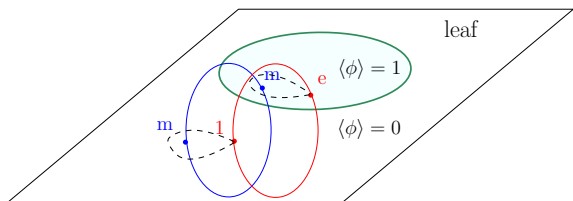

Figure 1: The three loop braiding process described by the correlation function (2.40), for $N = N' = 2$. $\phi = 0, 1$ (white and light green) label the two vacua on the leaves of the $\mathbb{Z}_2$ two-form gauge theory on the leaves, separated by a domain wall excitation (green string) which is confined to the leaves. The particles $e, m$ are the electric and magnetic particles of the $\mathbb{Z}_2$ one-form gauge theory on the leaves. The blue string consists of an $m$ particle everywhere it intersects a leaf. The red string is a string that consists of an $e$ particle excitation in the region $\phi = 1$ and trivial particles along the rest of the loop where $\phi = 0$ (if the domain wall is only on a single leaf as drawn in the figure). The three-loop braiding comes from the braiding between $e, m$ in the region $\phi = 1$.

## 2.7   Relation with higher-rank tensor gauge field

Let us relate foliated gauge fields discussed here to hollow (*i.e.* off-diagonal) symmetric rank two gauge field, which is used in the several effective field theories in the literature, *e.g.* [17, 22, 48, 49][15] for models with excitations of restricted mobility. The discussion is similar but different from that in [54].

Consider a hollow (*i.e.* off-diagonal) symmetric rank-two tensor gauge field in $d \geq 4$ spacetime dimensions

$$(a_0, \quad a_{ij}), \quad a_{ii} = 0 , \tag{2.42}$$

with the gauge transformation

$$a_0 \to a_0 + \partial_0 \lambda, \quad a_{ij} \to a_{ij} + \partial_i \partial_j \lambda, \quad \lambda \sim \lambda + 2\pi . \tag{2.43}$$

We now repackage it into

$$\mathcal{A}_0^k = \partial_k a_0, \quad \mathcal{A}_i^k = a_{ki} , \tag{2.44}$$

with the ordinary gauge transformation

$$\mathcal{A}^k \to \mathcal{A}^k + d\lambda^k, \quad \lambda^k \equiv \partial_k \lambda . \tag{2.45}$$

Note that the gauge parameter $\lambda^k$ can have a delta function in the $k$th direction.

The gauge field $\mathcal{A}^k$ has dimension two instead of one. Thus we can define the two-form gauge field

$$B_2^k = \mathcal{A}^k e^k , \tag{2.46}$$

which has the gauge transformation

$$B_2^k \to B_2^k + d\lambda_1^k, \quad \lambda_1^k = \lambda^k e^k . \tag{2.47}$$

Note the possible singularity in $\lambda_1^k$ from $\lambda^k$ is consistent with the discussion in Section 2.1. The two-form gauge field $B_2^k$ is a foliated two-form gauge field, which satisfies the constraint $B_2^k e^k = 0$. For simplicity, we can take the foliation one-forms to be $e^k = dx^k$ with spatial indices $k = 1, \cdots d - 1$.

---

[15]Examples of field theories with non-hollow symmetric tensor gauge fields are discussed in *e.g.* [50–53].

Let us verify the Dirac quantization condition of $B_2^k$, *i.e.* $\oint dB_2^k \in 2\pi\mathbb{Z}$ on three-spheres. Since the integral can be obtained by gluing two three-disks along the equator two-sphere, we only need to examine whether $\oint d\lambda_1^k$ is a multiple of $2\pi$ on two-sphere. The integral can be obtained by gluing two hemispheres, and thus it amounts to computing an integral on the equator of the two-sphere

$$\oint e^k \partial_k \lambda \, . \tag{2.48}$$

The loop integral can again be obtained by gluing two coordinate patches, across which $\lambda$ can jump by a multiple of $2\pi$. Thus we find $\oint d\lambda_1^k \in 2\pi$ (with potential discontinuities), and $B^k$ satisfies the Dirac quantization condition.

Note that the symmetric tensor gauge field $(a_0, a_{ij})$ does not correspond to the most general field configuration of $B_2^k$. For instance, if the gauge fields are dynamical with kinetic terms included in the action, the two theories have different dispersion relations. More precisely, the symmetric tensor gauge fields corresponds to foliated two-form gauge field with *flat* two-form gauge field

$$B := \sum_k B_2^k = \sum_{k,i} B_{ki}^k dx^k dx^i = \frac{1}{2} \sum_{k,i} \left( B_{ki}^k - B_{ik}^i \right) dx^k dx^i \, , \tag{2.49}$$

where we choose the $(d-1)$ foliation one-forms to be $e^k = dx^k$ with $k = 1, 2, \cdots d-1$ for the spatial directions, and $B_2^k = \sum_i B_{ki}^k dx^k dx^i$. Thus locally $B = d\Lambda$ where we can choose the gauge $\Lambda_k = 0$, then $\Lambda_0 = a_0$ (for $e^k = dx^k$), $B_{ki}^k = B_{ik}^i$ *i.e.* $a_{ik} = a_{ki}$. This shows that the two kinds of fields will result in different physics for gapless free field models, but the same physics for certain gapped models.

# 3 Twisted $\mathbb{Z}_N$ foliated two-form gauge theory in 3+1d

Consider the theory

$$\sum_{k,l} \frac{N p_{kl}}{4\pi} B_2^k B_2^l + \sum_k \frac{N}{2\pi} dA_1^k B_2^k \, , \tag{3.1}$$

where $B_2^k e^k = 0$. The gauge transformations are

$$B_2^k \to B_2^k + d\lambda_1^k, \quad A_1^k \to A_1^k + d\lambda_0^k + \alpha^k - \sum_l p_{kl} \lambda_1^l \, , \tag{3.2}$$

where $\alpha^k e^k = 0$, $\lambda_1^k e^k = 0$ and $d\lambda_0^k e^k = 0$. Since $B_2^k B_2^k = 0$, we set $p_{kk} = 0$. The equation of motion gives $N \sum_l p_{kl} B^l + N dA_1^k = 0$, $N dB_2^k = 0$.

We remark that the version of the theory with ordinary (*i.e.* not foliated) one-form and two-form gauge fields is studied in *e.g.* [33–35, 55], and it describes the low energy theory of suitable Walker-Wang model with boundary Abelian anyons [38, 56]. Here we will focus on the bulk property of the foliated model. The boundary properties will be investigated elsewhere.

One way to understand the theory to take $e^k = dx^k$, and express $B_2^k = u_1^k dx^k$. Then the action $\sum_{k,l} \frac{p_{kl}}{4\pi} B^k B^l = \sum_{k,l} \frac{p_{kl}}{4\pi} u_1^k u_1^l dx^k dx^l$ modifies the theory by inserting layers of surface operators $\oint \frac{p_{kl}}{4\pi} u_1^k u_1^l$.

## 3.1 Global symmetry

Let us couple the theory to a two-form background gauge field $C_E^k$ and a three-form background gauge field $C_M^k$

$$\frac{N}{2\pi} \sum_k B_2^k C_E^k + \frac{N}{2\pi} \sum_k A_1^k C_M^k \, . \tag{3.3}$$

The gauge field $C_M^k$ satisfies $C_M^k e^k = 0$, and they have the background gauge transformation $C_E^k \to C_E^k + d\lambda_1^k + \alpha_2^k$, $C_M^k \to C_M^k + d\lambda_2^k$ with $\alpha_2^k e^k = 0$ and $\lambda_2^k e^k = 0$.

In particular, the magnetic symmetry transforms $B_2^k \to B_2^k - \lambda_2^k$. Due to the coupling $\sum_{kl}(Np_{kl}/4\pi)B_2^k B_2^l$, part of the magnetic symmetry is broken explicitly, and the corresponding background is forced to be trivial. To see this, we use the equations of motion $\sum_l Np_{kl}B_2^l + NdA^k + NC_E^k = 0$, $NdB_2^k + NC_M^k = 0$, which implies $dC_E^k = \sum_l p_{kl}C_M^l$. For instance, if $p_{xy} = 1$, this implies that the gauge field $C_M^k$ is forced to be trivial and that the corresponding symmetry is broken by the topological term $p_{kl}$. In general, the symmetry corresponding to $C_M^k$ is $\mathbb{Z}_{r_k}$ with $r_k = \gcd_l(p_{kl}, N) := \gcd(p_{k1}, \cdots, p_{k,k-1}, p_{k,k+1}, \cdots, p_{k,n_k}, N)$, where the gcd is taken over all $l$.

## 3.2 Observables

The theory has operators

$$U_k = e^{i\int_\Sigma B_2^k}, \quad V_k = e^{i\oint_{\partial\Sigma} A_1^k + i\sum_l p_{kl}\int_\Sigma B_2^l}. \tag{3.4}$$

Both $U_k$ and $V_k$ are surface operators with boundary $\partial\Sigma$, where each connected component of $\partial\Sigma$ must lie on a leaf of the foliation $k$. The operators $U_k, V_k$ have $e^{2\pi i/N}$ statistics.

The operator $V_k$ is not a genuine line operator for $p_{kl} \neq 0$. The genuine line operators that describe a planons are integer powers of

$$V_k^{K_k} = e^{iK\oint_{\partial\Sigma} A_1^k + iK_k\sum_l p_{kl}\int_\Sigma B_2^l}, \tag{3.5}$$

where $K_k = N/L_k$ with $L_k = \gcd(p_{kl}, N)$, where the greatest common divisor is respect to all $l$. The part $\sum_l K_k p_{kl}\int_\Sigma B_2^l$ is trivial since it is a multiple of $N\int B_2^l$, and thus the operator $V^K$ only depends on the line $\partial\Sigma$, and it is a genuine line operator on the leaf *i.e.* describing a planon. The genuine line operators $V_k^K$ and $U_k$ have braiding $e^{2\pi i/L_k}$.

Let us consider the line operator

$$W_k = e^{i\oint\sum_k q_k A_1^k}. \tag{3.6}$$

Invariance under the gauge transformation $A_1^k \to A_1^k - \sum_l p_{kl}\lambda_1^l$ constrains the line operator to lie on the intersection of a leaf from each foliation $l$ that satisfies $\sum_k q_k p_{kl} \neq 0$ mod $N$. If three or more foliations satisfy that constraint, then the line operator describes a fracton. Else if only one or two foliations satisfy $\sum_k q_k p_{kl} \neq 0$ mod $N$, then the line operator describes a planon or lineon, respectively.

In summary, $e^{i\oint\sum_k q_k A^k}$ describes a particle with the mobility class of

- A planon, if only one of $q_k$ is not a multiple of $N$, denoted by $k = k_m$, and $q_{k_m}p_{k_m l} \in N\mathbb{Z}$ for $l \neq k_m$.

  Thus $q_{k_m}$ is a multiple of $\frac{N}{\gcd_{l \neq k_m}(p_{k_m l}, N)}$, where $\gcd_{l \neq k_m}(p_{k_m l}, N)$ is the greatest common divisor of $N$ and $p_{k_m l}$ with all $l \neq k_m$. There are $\gcd_{l \neq k_m}(p_{k_m l}, N)$ planons.

- A lineon, if at least one of $q_k$ is a multiple of $N$, denote such $k$ by $k = k_\star$, and $\sum_k q_k p_{kk_\star} \in N\mathbb{Z}$.

  If there are two $k_\star, k_\star'$ such that $q_{k_\star}, q_{k_\star'}$ are a multiple of $N$, then for $k_m \neq k_\star, k_\star'$, either (1) $q_{k_m}p_{k_m k_\star} \notin N\mathbb{Z}$ and $q_{k_m}p_{k_m k_\star'} \in N\mathbb{Z}$, or (2) $q_{k_m}p_{k_m k_\star'} \notin N\mathbb{Z}$ and $q_{k_m}p_{k_m k_\star} \in N\mathbb{Z}$.

  Suppose only one of $q_{k_\star}$ is a multiple of $N$. Then $q_{k_1}, q_{k_2}$ must be a multiple of $\frac{N}{\gcd(N, p_{k_1 k_\star}, p_{k_2 k_\star})}$. There are thus $\gcd(N, p_{k_1 k_\star}, p_{k_2 k_\star}) - 2$ of them. Suppose $q_{k_\star}, q_{k_\star'}$ are both multiples of $N$. In case (1) this means $q_{k_m} \notin \frac{N}{\gcd(N, p_{k_m k_\star})}\mathbb{Z}$ and $q_{k_m} \in \frac{N}{\gcd(N, p_{k_m k_\star'})}\mathbb{Z}$. Denote

$q_{k_m} = m \frac{N}{\gcd(N, p_{k_m k'_\star})}$ with $m \sim m + \gcd(N, p_{k_m k_\star})$, then $m \gcd(N, p_{k_m k_\star}) \notin \gcd(N, p_{k_m k'_\star})\mathbb{Z}$,

i.e. $m \notin \frac{\gcd(N, p_{k_m k'_\star})}{\gcd(N, p_{k_m k_\star}, p_{k_m k'_\star})}\mathbb{Z}$. Thus there are $\gcd(N, p_{k_m k'_\star}) - \gcd(N, p_{k_m k_\star}, p_{k_m k'_\star})$ of them. Case (2) is obtained by exchanging $k_\star$ with $k'_\star$. Some of them can be obtained by fusing planons.

- A fracton, if $\sum_k q_k p_{kl} \notin N\mathbb{Z}$ for all three $l$ in 3+1d.

  This requires at least one $q_k$ satisfies $q_k \notin \frac{N}{\gcd_l(N, p_{kl})}\mathbb{Z}$ for all $l \neq k$. Then such $k$ contributes $N - \sum_{l \neq k} \gcd_l(N, p_{kl})$ fractons.

## 3.3 Correlation function

The correlation function of the non-foliation version of the theory is computed in Section 7 of [47]. The discussion of the foliation version is essentially the same, except that there are more gauge invariant operators when the support of the operator is suitably chosen, such as on the intersection of leaves of foliations.

For instance, let us compute the correlation function for the planon $e^{i \frac{N}{\gcd_l(p_{kl}, N)} \oint_{\gamma_k} A^k}$, where we insert the operator at closed loop $\gamma_k$ on a leaf foliation $k$. This amounts to deforming the action with

$$\frac{N}{2\pi}\left(p_{12}B^1 B^2 + p_{13}B^1 B^3 + p_{23}B^2 B^3\right) + \sum_k \frac{N}{2\pi}B^k dA^k + \frac{N}{\gcd_l(p_{kl}, N)}A^k \delta(\gamma_k)^\perp, \qquad (3.7)$$

where $\delta(\gamma_k)^\perp$ is the delta function three-form that restricts the spacetime integral to $\gamma_k$. Integrating out $A^k$ gives

$$dB^k = -\frac{2\pi}{\gcd_l(p_{kl}, N)}\delta(\gamma_k)^\perp. \qquad (3.8)$$

On $\mathbb{R}^4$ it can be solved using surface $\Sigma_k$ with boundary $\gamma_k$ as

$$B^k = -\frac{2\pi}{\gcd_l(p_{kl}, N)}\delta(\Sigma_k)^\perp. \qquad (3.9)$$

Then substituting into the remaining action gives a trivial correlation function for these planons; *i.e.* they have trivial self statistics and mutual statistics,

$$\langle \prod_k e^{i \frac{N}{\gcd_l(p_{kl}, N)} \oint_{\gamma_k} A^k}\rangle = \exp\left(\frac{2\pi i N p_{12}}{\gcd(p_{12}, p_{13}, N)\gcd(p_{12}, p_{23}, N)}\delta(\Sigma_1)^\perp \delta(\Sigma_2)^\perp\right)$$

$$\cdot \exp\left(\frac{2\pi i N p_{13}}{\gcd(p_{12}, p_{13}, N)\gcd(p_{13}, p_{23}, N)}\delta(\Sigma_1)^\perp \delta(\Sigma_3)^\perp\right)$$

$$\cdot \exp\left(\frac{2\pi i N p_{23}}{\gcd(p_{12}, p_{23}, N)\gcd(p_{13}, p_{23}, N)}\delta(\Sigma_2)^\perp \delta(\Sigma_3)^\perp\right) = 1. \qquad (3.10)$$

We can also consider the correlation function of the planon $e^{i \oint_{\gamma_k} A^k + \sum_l \int_{\Sigma_k} p_{kl} B^l}$, where $\Sigma_{kl}$ has boundary $\gamma_k$. Repeating the above steps, we find these planons have mutual statistics:

$$\exp -\frac{2\pi i}{N}\left(p_{12}\delta(\Sigma_1)^\perp \delta(\Sigma_2)^\perp + p_{13}\delta(\Sigma_1)^\perp \delta(\Sigma_3)^\perp + p_{23}\delta(\Sigma_2)^\perp \delta(\Sigma_3)^\perp\right). \qquad (3.11)$$

In other words, for the basic planons on the leaves of foliation $i, j$, they have mutual statistics $e^{-2\pi i p_{ij}/N}$.

Consider the lineon $e^{i\oint \eta^{31}A^1 - \eta^{32}A^2}$, where $\eta, \kappa$ are given in Figure 5,

$$d_{ij} = \gcd(p_{ij}, N), \quad \overline{p}_{ij}p_{ij} = d_{ij} \bmod N, \quad \eta_{ij} = \overline{p}_{ij}\frac{\kappa_i}{d_{ij}},$$

$$\kappa_i = \text{lcm}(d_{ij}, d_{ik}) \text{ for distinct } i, j, k. \tag{3.12}$$

The correlation function of the lineon can be computed in a similar way[16]

$$\langle e^{i\oint_{\gamma_{12}} \eta^{31}A^1 - \eta^{32}A^2} e^{i\oint_{\gamma_{13}} \eta^{21}A^1 - \eta^{23}A^3} e^{i\oint_{\gamma_{23}} \eta^{12}A^2 - \eta^{13}A^3} \rangle$$

$$= \exp -\frac{2\pi i}{N} \left( p_{12}\eta_{31}\eta_{32}\delta(\Sigma_{12})^{\perp}\delta(\widetilde{\Sigma}_{12})^{\perp} + p_{13}\eta_{21}\eta_{23}\delta(\Sigma_{13})^{\perp}\delta(\widetilde{\Sigma}_{13})^{\perp} \right.$$

$$\left. + p_{23}\eta_{12}\eta_{13}\delta(\Sigma_{23})^{\perp}\delta(\widetilde{\Sigma}_{23})^{\perp} \right)$$

$$\cdot \exp \frac{2\pi i}{N} \left( p_{12}\eta^{31} \right) \tag{3.14}$$

where $\Sigma_{ij}, \widetilde{\Sigma}_{ij}$ are surfaces with boundary $\gamma_{ij}$, and they can be thought of as related by pushoff along some framing direction.

## 3.4 Exactly solvable Hamiltonian model

Let us construct a Hamiltonian model using a similar method as in [36] to construct a lattice Hamiltonian model for the one-form symmetry SPT phase. The model for the two-form gauge theory then follows from gauging the symmetry.

### 3.4.1 SPT phase with subsystem symmetry

Let us consider a cubic lattice and choose $e^1 = dx, e^2 = dy, e^3 = dz$. After integrating out $A_1^k$, which forces $B_2^k$ to be $\mathbb{Z}_N$ two-form foliated gauge field, the action can be expressed in the discrete notation as $\frac{2\pi}{N}\phi_4(B^1, B^2, B^3)$ with (we normalize $B^k = 0, 1, \cdots, N-1 \bmod N$):[17]

$$\phi_4(B^1, B^2, B^3) = p_{12}B^1 \cup B^2 + p_{13}B^1 \cup B^3 + p_{23}B^2 \cup B^3. \tag{3.15}$$

We have

$$\phi_4(d\lambda^1, d\lambda^2, d\lambda^3) = d\phi_3(\lambda^1, \lambda^2, \lambda^3), \tag{3.16}$$

where

$$\phi_3(\lambda^1, \lambda^2, \lambda^3) = p_{12}\lambda^1 \cup d\lambda^2 + p_{13}\lambda^1 \cup d\lambda^3 + p_{23}\lambda^2 \cup d\lambda^3. \tag{3.17}$$

A Hamiltonian model for the SPT phase with symmetry $\lambda^i \to \lambda^i + s^i$ is given by conjugating the Ising paramagnet $H^0 = -\sum_{n=0}^{N-1} \left( \sum_{e_x} X_{e_x}^n + \sum X_{e_y}^n + \sum X_{e_z}^n \right)$ by $e^{(2\pi i/N)\int \phi_3}$, where $Z, X$ are the $\mathbb{Z}_N$ clock and shift Pauli matrices satisfying $X_e^{\dagger}Z_e X_e = e^{2\pi i/N}Z_e$:

$$H_{\text{SPT}} = -\sum_{n=0}^{N-1} \left( \sum X_{e_x}^n e^{\frac{2\pi i}{N}\int \phi_3(\lambda^1 + n\widetilde{e}_x, \lambda^2, \lambda^3) - \phi_3(\lambda^1, \lambda^2, \lambda^3)} \right.$$

$$+ \sum X_{e_y}^n e^{\frac{2\pi i}{N}\int \phi_3(\lambda^1, \lambda^2 + n\widetilde{e}_y, \lambda^3) - \phi_3(\lambda^1, \lambda^2, \lambda^3)}$$

$$\left. + \sum X_{e_z}^n e^{\frac{2\pi i}{N}\int \phi_3(\lambda^1, \lambda^2, \lambda^3 + n\widetilde{e}_z) - \phi_3(\lambda^1, \lambda^2, \lambda^3)} \right), \tag{3.18}$$

---

[16]The equation of motion gives

$$B^1 = \eta_{31}\delta(\Sigma_{12}) + \eta_{21}\delta(\Sigma_{13}), \quad B^2 = -\eta_{32}\delta(\widetilde{\Sigma}_{12}) + \eta_{12}\delta(\Sigma_{23}), \quad B^3 = -\eta_{23}\delta(\widetilde{\Sigma}_{13}) - \eta_{13}\delta(\widetilde{\Sigma}_{23}). \tag{3.13}$$

[17]For a review of cup product, see *e.g.* [36, 57–59]

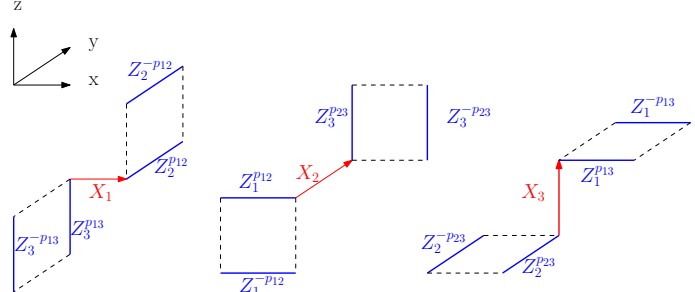

Figure 2: The Hamiltonian model (3.19) for the subsystem SPT phase, where there are $x^k$-type $\mathbb{Z}_N$ degrees of freedom on the edges in the $x^k$ direction, acted upon by Pauli matrices $X_k, Y_k, Z_k$. One subsystem symmetry is a product of $X$ operators over edges in the $x$ direction that intersect a $yz$ plane on the dual lattice. Subsystem symmetries for $xz$ and $xy$ planes are similar.

where $\widetilde{e}_x$ is the one-cochain that takes value 1 on the edge $e_x$ in the $x$ direction and 0 otherwise, and similarly for $\widetilde{e}_y, \widetilde{e}_z$. Explicitly,

$$H_{\text{SPT}} = -\sum_{n=0}^{N-1} \Big( \sum X_{e_x}^n e^{\frac{2\pi in}{N} \int p_{12} \widetilde{e}_x \cup d\lambda^2 + p_{13} \widetilde{e}_x \cup d\lambda^3}$$
$$+ \sum X_{e_y}^n e^{\frac{2\pi in}{N} \int p_{12} d\lambda^1 \cup \widetilde{e}_y + p_{23} \widetilde{e}_y \cup d\lambda^3}$$
$$+ \sum X_{e_z}^n e^{\frac{2\pi in}{N} \int p_{13} d\lambda^1 \cup \widetilde{e}_z + p_{23} d\lambda^2 \cup \widetilde{e}_z} \Big) . \tag{3.19}$$

The Hamiltonian model is in Figure 2.

### 3.4.2 Gauged SSPT phase: two-form gauge theory

Next, we gauge the symmetry by introducing a gauge field described by $\mathbb{Z}_N$ degrees of freedom on each face, and impose the Gauss constraint

$$X_e \prod X_f^o = 1 , \tag{3.20}$$

where the product is over the faces adjacent to the edge $e$, and $o = \pm 1$ depends on the orientation of $f$ relative to $e$. We can choose a branching structure such that on the $2d$ plane orthogonal to $e$ with $e$ pointing into the plane, the product is over two $X_f^{-1}$ on the upper and left faces and two $X_f$ on the lower and right faces. For the face that shares an edge in the $k$ direction, we associate a two-form gauge field $B^k = 0, 1, \cdots N-1$, where $Z^k$ has eigenvalue $e^{(2\pi i/N)B^k}$. For the Hamiltonian to commute with the gauge constraint, we couple each term to gauge field $B^k$ and replace $d\lambda^k$ with $d\lambda^k + B^k$. We also include a flux term to impose the condition $dB^k = 0 \mod N$ on the ground state. Then we use the Gauss constraint to gauge-fix $\lambda^k = 0$. We obtain a Hamiltonian for the theory after gauging the symmetry:

$$H_{\text{gauged}} = -\sum_{n=0}^{N-1} \Big( \sum \big( \prod X_{f_x}^o \big)^n e^{\frac{2\pi in}{N} \int p_{12} \widetilde{e}_x \cup B^2 + p_{13} \widetilde{e}_x \cup B^3}$$
$$+ \sum \big( \prod X_{f_y}^o \big)^n e^{\frac{2\pi in}{N} \int p_{12} B^1 \cup \widetilde{e}_y + p_{23} \widetilde{e}_y \cup B^3}$$
$$+ \sum \big( \prod X_{f_z}^o \big)^n e^{\frac{2\pi in}{N} \int p_{13} B^1 \cup \widetilde{e}_z + p_{23} B^2 \cup \widetilde{e}_z}$$
$$+ \sum_c \big( \prod Z_f^o \big)^n \Big) . \tag{3.21}$$

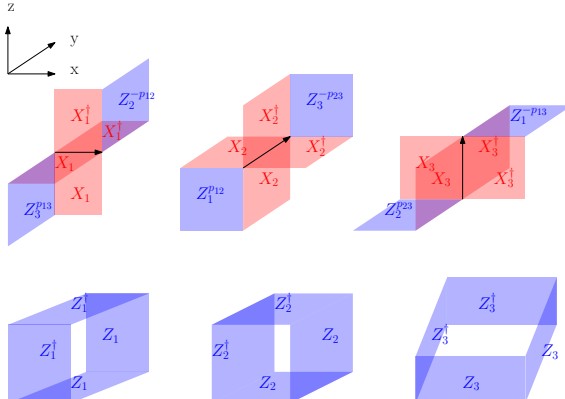

Figure 3: Hamiltonian model for the twisted foliated $\mathbb{Z}_N$ gauge theory. We label the top row of excitations as $e^{YZ}, e^{XZ}, e^{XY}$, and the bottom row as $f^1, f^2, f^3$ (from left to right). Each face has two $\mathbb{Z}_N$ degrees of freedom: a $x^k - x^l$ planar face has an $x^k$-type and $x^l$-type $\mathbb{Z}_N$ degree of freedom.

The edge and flux terms are given by summing over $n = 0, \cdots N - 1$ powers of the terms in Figure 3.[18]

Consider general $N$ with prime factorization $N = \prod_i q_i^{r_i}$ where each $q_i$ is prime. The ground state degeneracy can be obtained by the method of [60, 61]:[19]

$$\text{GSD} = \prod_i q_i^{2r_i(l_x + l_y + l_z) - c_i} \tag{3.23}$$

$$c_i = 2\max(b_{12}^{(i)}, b_{23}^{(i)}, b_{13}^{(i)}) + \text{median}(b_{12}^{(i)}, b_{23}^{(i)}, b_{13}^{(i)})$$

$$b_{kl}^{(i)} = r_i - \log_{q_i} \gcd(q_i^{r_i}, p_{kl}).$$

When $p_{kl} = 0$ for all $k, l$, this reproduces the ground state degeneracy of the theory in Section 2.5.

As a consistency check, we note that if $N = mn$ for $m, n$ coprime, a foliated $\mathbb{Z}_N$ two-form gauge field can be decomposed uniquely into foliated $\mathbb{Z}_m$ and $\mathbb{Z}_n$ two-form gauge fields as $B^k = mB^k_{\mathbb{Z}_n} + nB^k_{\mathbb{Z}_m}$, where the normalization is $B^k = 0, 1 \cdots N - 1$, $B^k_{\mathbb{Z}_n} = 0, 1 \cdots, n - 1$ and $B^k_{\mathbb{Z}_m} = 0, 1 \cdots, m - 1$. Then the action $e^{iS}$ of the $\mathbb{Z}_N$ foliated two-form gauge field is (after integrating out gauge field $A_1^k$)

$$\exp\left(\frac{\pi i}{N} \sum_{k,l} p_{kl} B^k B^l\right) = \exp\left(\frac{\pi i}{n} \sum_{k,l} mp_{kl} B^k_{\mathbb{Z}_n} B^l_{\mathbb{Z}_n}\right) \exp\left(\frac{\pi i}{m} \sum_{k,l} np_{kl} B^k_{\mathbb{Z}_m} B^l_{\mathbb{Z}_m}\right). \tag{3.24}$$

Thus the theory of $(N, p_{kl})$ factorizes into the product of the theory of $(n, mp_{kl})$ and $(m, np_{kl})$. This is consistent with the ground state degeneracy in (3.23).

**Excitations** We tabulate the kinds of excitations of Figure 3 in Table 1. An excitation of the top row is a planon, which we label $e^{YZ}, e^{XZ}, e^{XY}$ (from left to right). These planons can be

---

[18]The model has a $\mathbb{Z}_3$ rotation symmetry about the (1,1,1) direction given by:

$$x \to y \to z \to x$$
$$X_1 \to X_2 \to X_3 \to X_1$$
$$Z_1 \to Z_2 \to Z_3 \to Z_1. \tag{3.22}$$

[19]See also Appendix B of [62] for a brief review. We used the computer code in [63].

Table 1: Excitations of Figure 3 for $p_{kl} \not\equiv 0 \pmod{N}$.

| Excitations | Mobility |
|---|---|
| $e^{YZ}$ | YZ-planon |
| $e^{XZ}$ | XZ-planon |
| $e^{XY}$ | XY-planon |
| $f^1$ or $f^2$ or $f^3$ | fracton |
| $f_0^1 (f_{\hat{x}}^1)^{-1}$ | YZ-planon |
| $f_0^2 (f_{\hat{y}}^2)^{-1}$ | XZ-planon |
| $f_0^3 (f_{\hat{z}}^3)^{-1}$ | XY-planon |
| $(f_{\hat{x}}^2)^{\eta_{1,2}} (f_0^3)^{\eta_{1,3}}$ | X-lineon |
| $(f_{\hat{y}}^3)^{\eta_{2,3}} (f_0^1)^{\eta_{2,1}}$ | Y-lineon |
| $(f_{\hat{z}}^1)^{\eta_{3,1}} (f_0^2)^{\eta_{3,2}}$ | Z-lineon |

moved via the unitary action of $Z$ operators. When $p_{kl} \not\equiv 0 \pmod{N}$, which we will assume for the remainder of this section[20], any excitation of the bottom row is a fracton, which we label $f^1, f^2, f^3$ (from left to right). A dipole of $f^1$ fractons displaced in the x-direction is a YZ-planon; it can move along a YZ plane using the operators shown in Figure 4.[21] We denote this dipole as $f_0^1 (f_{\hat{x}}^1)^{-1}$, where the subscript denotes the lattice position of the excited operator. Combinations of fractons can also result in a lineon. For example, an X-lineon results from $\eta_{1,3}$ many $f^3$ fractons along with $\eta_{1,2}$ many $f^2$ fractons displaced in the x-direction, where

$$\eta_{i,j} p_{i,j} \equiv \eta_{i,k} p_{i,k} \bmod N \ , \tag{3.25}$$

for any three distinct indices $i, j, k \in \{1, 2, 3\}$. We denote this composite as $(f_{\hat{x}}^2)^{\eta_{1,2}} (f_0^3)^{\eta_{1,3}}$ in the table. These lineons can be moved via the operators shown in Figure 5.

**Fusion rules** We can determine a basis of fusion rules (which can be formalized using a module [10]) for the excitations by asking which combination of excitations can fuse to the vacuum. In other words, we ask what excitations can be created or annihilated by local operators, such as a single $X$ or $Z$ operator, when acting on the ground state.[22] The resulting fusion rules are shown in Table 2. These fusion rules (along with the other two sets generated by symmetry (3.22)) are a basis that generates all possible fusion rules. From the first two rows in Table 2, we see the fractonic behavior of $f_0$, as moving it requires creating a planon $e^{(XY)}$. In contrast, the last two rows show that $e^{(YZ)}$ can move along a YZ plane.

### 3.4.3 Two-form gauge theory with $N = 2, p = 1$: variant of X-cube model

Consider the example $N = 2$, $p_{12} = p_{21} = p_{23} = p_{32} = p_{13} = p_{31} = 1$. The Hamiltonian is given in Figure 6. The ground state degeneracy on a three-torus with lengths $l_x, l_y, l_z$ is the

---

[20]Let $(f^k)^n$ denote the fusion of $n$ many $f^k$, with $n \not\equiv 0 \pmod{N}$. If $np_{12} \equiv np_{23} \equiv np_{13} \equiv 0 \pmod{N}$, then the three $(f^k)^n$ for $k = 1, 2, 3$ are all planons. If $np_{12} \not\equiv 0$ while $np_{23} \equiv np_{13} \equiv 0 \pmod{N}$, then $(f^1)^n$ and $(f^2)^n$ are lineons while $(f^3)^n$ is a planon. If $np_{12} \not\equiv 0$ and $np_{23} \not\equiv 0$ while $p_{13} \equiv 0 \pmod{N}$, then $(f^1)^n$ and $(f^3)^n$ are lineons while $(f^2)^n$ is a fracton. Thus, we see that each $np_{kl} \not\equiv 0 \pmod{N}$ restricts the mobility of $(f^k)^n$ and $(f^l)^n$ by one dimension each.

[21]Strings of the operators in Figure 4 can be used to move the planons anywhere within their plane, analogous to how string operators move the toric code excitations around.

[22]The superselection sectors are obtained by modding out the set of all possible excitations by the set of trivial excitations (*i.e.* excitations created or annihilated by local operators) which fuse to the vacuum.

Table 2: Excitations of Figure 3 that can be fused to the vacuum. We only list the excitations annihilated by acting with $X_1$ or $Z_1$, since the others can be obtained by symmetry (3.22). In the second column, $X_1^{(XY)}$ is an $X_1$ operator acting on a XY-plaquette, and similar for the other operators.

| Excitations that fuse to vacuum | Annihilated by |
|---|---|
| $(f_0^1)^{-1} f_{\widehat{z}}^1 \left( e_{-\frac{1}{2}\widehat{x}-\frac{1}{2}\widehat{y}-\frac{1}{2}\widehat{z}}^{(XY)} \right)^{-p_{13}} \to 1$ | $X_1^{(XY)}$ |
| $f_0^1 (f_{\widehat{y}}^1)^{-1} \left( e_{\frac{1}{2}\widehat{x}+\frac{1}{2}\widehat{y}+\frac{1}{2}\widehat{z}}^{(XZ)} \right)^{p_{12}} \to 1$ | $X_1^{(XZ)}$ |
| $e_0^{(YZ)} (e_{\widehat{y}}^{(YZ)})^{-1} \to 1$ | $Z_1^{(XY)}$ |
| $e_0^{(YZ)} (e_{\widehat{z}}^{(YZ)})^{-1} \to 1$ | $Z_1^{(XZ)}$ |

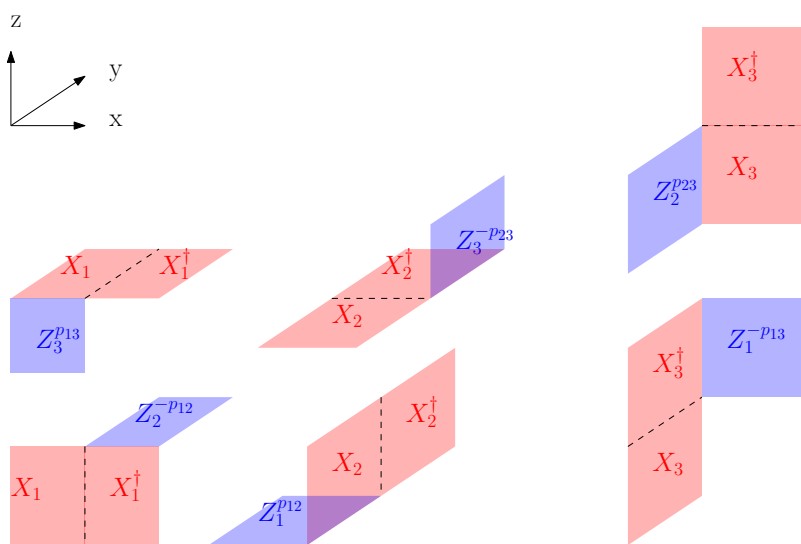

Figure 4: Operators creating planons (as dipoles of $f^i$ excitations) that commute with the edge terms. They have non-trivial mutual commutation relations.

same as the X-cube model:

$$\text{GSD} = 2^{2l_x + 2l_y + 2l_z - 3} . \tag{3.26}$$

Our model also has the same fusion module [10] as the X-cube model. That is, the excitations can be mapped to X-cube excitations with the same fusion. Our fractons $f^i$ are mapped as follows:

$$
\begin{aligned}
f_0^1 &\longleftrightarrow F_{\widehat{y}} L_{-\frac{1}{2}\widehat{x}+\frac{1}{2}\widehat{y}-\frac{1}{2}\widehat{z}}^1 \\
f_0^2 &\longleftrightarrow F_{\widehat{z}} L_{-\frac{1}{2}\widehat{x}-\frac{1}{2}\widehat{y}+\frac{1}{2}\widehat{z}}^2 \\
f_0^3 &\longleftrightarrow F_{\widehat{x}} L_{+\frac{1}{2}\widehat{x}-\frac{1}{2}\widehat{y}-\frac{1}{2}\widehat{z}}^3 ,
\end{aligned}
\tag{3.27}
$$

where $F_r$ is an X-cube fracton centered at $r$, while $L_r^i$ is an $x^i$-axis X-cube lineon centered at $r$. Our planons $e$ can be mapped to X-cube particles by first mapping them a pair of fractons via the first two fusion rules in Table 2, and then applying the above mapping.

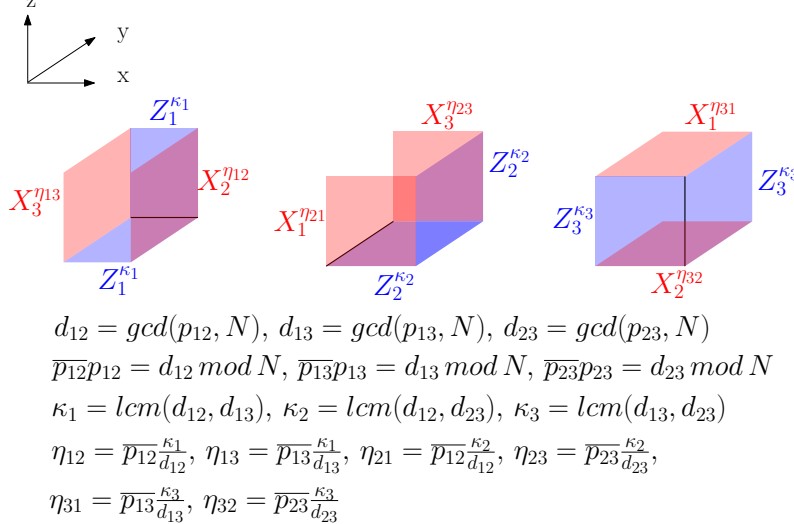

$$d_{12} = gcd(p_{12}, N),\ d_{13} = gcd(p_{13}, N),\ d_{23} = gcd(p_{23}, N)$$
$$\overline{p_{12}}p_{12} = d_{12}\,mod\,N,\ \overline{p_{13}}p_{13} = d_{13}\,mod\,N,\ \overline{p_{23}}p_{23} = d_{23}\,mod\,N$$
$$\kappa_1 = lcm(d_{12}, d_{13}),\ \kappa_2 = lcm(d_{12}, d_{23}),\ \kappa_3 = lcm(d_{13}, d_{23})$$
$$\eta_{12} = \overline{p_{12}}\tfrac{\kappa_1}{d_{12}},\ \eta_{13} = \overline{p_{13}}\tfrac{\kappa_1}{d_{13}},\ \eta_{21} = \overline{p_{12}}\tfrac{\kappa_2}{d_{12}},\ \eta_{23} = \overline{p_{23}}\tfrac{\kappa_2}{d_{23}},$$
$$\eta_{31} = \overline{p_{13}}\tfrac{\kappa_3}{d_{13}},\ \eta_{32} = \overline{p_{23}}\tfrac{\kappa_3}{d_{23}}$$

Figure 5: Operators creating lineons that commute with the edge terms. They satisfy non-trivial commutation relations. $\kappa_i = b_{ij}p_{ij}$ [see (3.25)] for any $j \neq i$ with $j \in \{1, 2, 3\}$. An explicit solution for $\eta$ is given below the figure.

**Inequivalence to X cube model**  Although the ground state degeneracy and the fusion module of the model with $N = 2, p_{kl} = 1$ is the same as the X-cube model, this model is *not* local unitary equivalent to the X-cube model.

This is consistent with the fact that in the mapping of fusion module (3.27), although our planons $e$ all commute with each other, this commutation relation is not preserved under the mapping.

To show that the two theories are inequivalent, suppose (for contradiction) that the ground states are equivalent to X-cube up to a local unitary transformation (and possible addition of decoupled qubits). Then the lineon excitations (bottom three rows of Table 1) must have the same quotient superselection sectors (QSS)[23] as the X-cube lineon excitations. In order for the QSS fusion to be consistent (*i.e.* pairs of fractons must fuse into lineons as in the last three rows of Table 1), the $f^i$ fractons must have the same QSS as an X-cube fracton fused with an X-cube $x^i$-axis lineon. Now note that if we act with e.g. a $X_3$ operator on a YZ-plane plaquette, then we create $f_0^3 f_{\hat{x}}^3 e_{\frac{1}{2}\hat{y}+\frac{1}{2}\hat{z}}^{YZ}$ from the vacuum; *i.e.* these particles fuse to the vacuum. This fusion implies that the $e$ planons must consist of at least two X-cube fractons[24]. Since the $e$ planons all commute with each other, this implies that the $e$ planons must not consist of any X-cube lineons. But that is impossible since $f_0^3$ has a QSS that contains an X-cube z-axis lineon, which implies that $f_0^3 f_{\hat{x}}^3$ must consist of some X-cube lineons, even though $f_0^3 f_{\hat{x}}^3$ must fuse to $e^{YZ}$, which doesn't contain any X-cube lineons.

The possible equivalence to twisted X-cube models, such as the semionic X-cube model [65], is left as an open question.

---

[23]The quotient superselection sector (QSS) [64] is the superselection sector that results from modding out by all planon superselection sectors. The X-cube model has $2^3 = 8$ different QSS, which are generated by the fracton, x-axis lineon, and y-axis lineon.

[24]In the X-cube model, the fracton charge is conserved mod 2 on each plane. Since $f^3$ consists of an odd number of X-cube fractons, $f^3$ must also have an odd charge on at least one YZ plane. Thus $f_0^3 f_{\hat{x}}^3$ must have an odd charge on at least two YZ planes. $e_{\frac{1}{2}\hat{y}+\frac{1}{2}\hat{z}}^{YZ}$ must also have an odd charge on the same two planes.

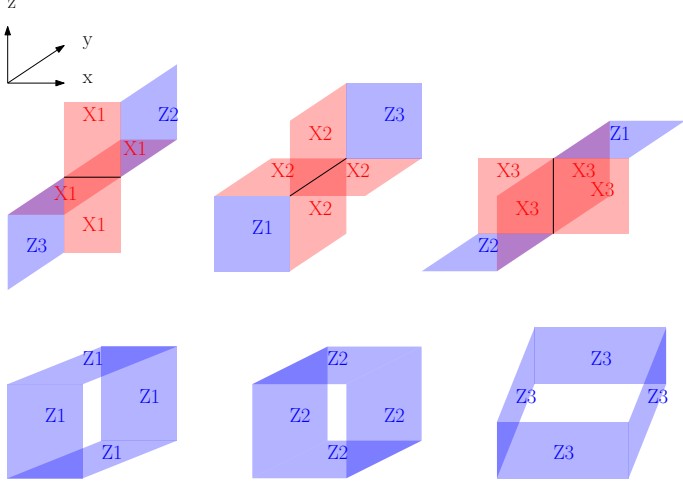

Figure 6: The edge and flux terms for the Hamiltonian model of twisted foliated $\mathbb{Z}_2$ two-form gauge theory. Each face in a $x^k - x^l$ plane has two qubits of types $x^k, x^l$. We label the top row of excitations as $e^{YZ}, e^{XZ}, e^{XY}$, and the bottom row as $f^1, f^2, f^3$ (from left to right).

**Twisted foliated two-form gauge theory**   Let us compare the lattice model with the twisted foliated two-form gauge theory,

$$\frac{2}{2\pi}\left(B^1 B^2 + B^1 B^3 + B^2 B^3\right) + \frac{2}{2\pi}\left(B^1 dA^1 + B^2 dA^2 + B^3 dA^3\right). \tag{3.28}$$

The gauge transformation is

$$
\begin{aligned}
&B^k \to B^k + d\lambda_1^k \\
&A^1 \to A^1 - \lambda_1^2 - \lambda_1^3 + d\lambda_0^1 + \alpha_1^1, \quad A^2 \to A^2 - \lambda_1^1 - \lambda_1^3 + d\lambda_0^2 + \alpha_1^2 \\
&A^3 \to A^3 - \lambda_1^1 - \lambda_1^1 + d\lambda_0^3 + \alpha_1^3.
\end{aligned}
\tag{3.29}
$$

The spectrum of operators is

- Fracton $e^{i\oint A^k}$ where the curve is supported on the intersection of a leaf for each foliation.

- Lineon $e^{i\oint A^k - A^l}$ with $k \neq l$.

- Planon $e^{i\oint A^1 + i\int B^2 + B^3}$ that describes a dipole of fractons, where the surface is a thin ribbon.

The spectrum of particles is similar to the X-cube model, but this fracton can fuse into lineons. Let us compute the correlation function of lineons. We insert the operators

$$\int_\gamma \left(A^1 - A^2\right) + \int_{\gamma'} \left(A^2 - A^3\right). \tag{3.30}$$

Then integrating out $A^k$ gives $B^1 = \pi\delta(\Sigma)^\perp$, $B^2 = -\pi\delta(\widetilde{\Sigma})^\perp + \pi\delta(\widetilde{\Sigma}')^\perp$, $B^3 = -\pi\delta(\Sigma')^\perp$ with $\partial\Sigma = \partial\widetilde{\Sigma} = \gamma$, $\partial\Sigma' = \partial\widetilde{\Sigma}' = \gamma'$. The surfaces $\widetilde{\Sigma}, \widetilde{\Sigma}'$ have the same boundary as $\Sigma, \Sigma'$, and they can be thought of as related by pushoff along some framing direction. Evaluating the rest of the action produces

$$\langle e^{i\oint_\gamma A^1 - A^2} e^{i\oint_{\gamma'} A^2 - A^3}\rangle = (-1)^{\int \delta(\Sigma)^\perp \delta(\widetilde{\Sigma})^\perp + \delta(\Sigma')^\perp \delta(\widetilde{\Sigma}')^\perp} \cdot (-1)^{\int \delta(\Sigma)^\perp \delta(\Sigma')^\perp + \delta(\widetilde{\Sigma})^\perp \delta(\Sigma')^\perp + \delta(\Sigma)^\perp \delta(\widetilde{\Sigma}')^\perp}. \tag{3.31}$$

Thus the lineons have $\pi$ self statistics and $\pi$ mutual statistics.

# 4 Electric model

Consider the theory

$$S_E = \frac{N}{2\pi}\sum_k dA^k B^k + \frac{N}{2\pi}\sum_k \eta_k(a)B^k + S_{\text{top}}(A^k, a)\,, \tag{4.1}$$

where $a$ is a $G$ gauge field for some finite or continuous group $G$, $\eta_k \in H^2(G, \mathbb{Z}_N)$, which can be ordinary or twisted cohomology if $G$ has permutation action on $\mathbb{Z}_N$, $B^k$ is a foliated two-form gauge field satisfying $B^k e^k = 0$, and $A^k$ is a one-form gauge field. The last term is the "symmetry twist"

$$S_{\text{top}}(A^k, a) = \omega(a) + \frac{N}{2\pi}\sum_k A^k \nu_k(a)\delta(\mathcal{L}_k)^\perp\,, \tag{4.2}$$

where the first term only depends on the $G$ gauge field by $\omega \in H^4(G, U(1))$ or some cobordism group generalization, $\nu_k \in H^2(G, \mathbb{Z}_N)$, and $\delta(\mathcal{L}_k)^\perp$ is the Poincaré dual of some leaf $\mathcal{L}_k$ of foliation $k$, which is included to make this term invariant under $A^k \to A^k + \alpha^k$ with $\alpha^k e^k = 0$. In other words, it is a topological term supported on a single leaf.

The model has the property that the equation of motion for $B^k$ imposes $dA^k + \eta_k(a) = 0$ on the leaf of foliation $k$, which relates the gauge fields $A^k$ and $a$, and thus the name "electric model". Later in Section 5 we will consider another set of models that relate $B^k, a$, which we call "magnetic models". More general models are a mixture of the two kinds.

**Symmetry fractionalization**   Let us first explore some kinematic conditions. Integrating out $B^k$ gives[25]

$$dA^k + \eta_k(a) = 0 \text{ on a leaf of foliation } k\,, \tag{4.3}$$

which implies $(A^k, a)$ describes a $H_k$ gauge field on a leaf of foliation $k$ given by the extension

$$1 \to \mathbb{Z}_N \to H_k \to G \to 1\,, \tag{4.4}$$

specified by $\eta_k \in H^2(G, \mathbb{Z}_N)$. This implies that the operator

$$e^{i\int A^k} \tag{4.5}$$

carries a projective representation specified by $\eta_k$ under $G$.

Similarly, suppose $L_k = 0$. Then integrating out $A^k$ implies that the operator

$$e^{i\int B^k} \tag{4.6}$$

carries $G$ symmetry fractionalization, *i.e.* a $G$ symmetry anomaly on the world volume described by $\nu_k\delta(\mathcal{L}_k)^\perp$. One way to interpret $\nu_k$ is that if we slice the loop created on the boundary of $e^{i\int B^k}$ by the leaf, then we will find a particle carrying a projective representation described by $\nu_k$.

**Anomaly and symmetry response**   The coupling for general $\eta$ and $\nu$ implies that the $G$ symmetry is anomalous. The anomaly is described by the SPT phase in one dimension higher with subsystem symmetry, whose effective action is

$$\frac{N}{2\pi}\int \sum_k \eta_k \nu_k(a)\delta(\mathcal{L}_k)^\perp\,. \tag{4.7}$$

---

[25]Since $B^k$ satisfies the constraint $B^k e^k = 0$, the equation of motion holds up to fields vanishing on a leaf of foliation $k$.

In some cases, it can be cancelled by a local term, in which case there is no anomaly. For instance, this is the case when $G = U(1)$, $\eta_k(a) = qda/N$, $\nu_k(a) = q'da/N$ for integers $q, q'$, and the anomaly can be cancelled by $\frac{qq'/N}{2\pi} ada\delta(\mathcal{L}_k)^{\perp}$. This implies that the leaf $\mathcal{L}_k$ of foliation $k$ has Hall conductance $\sigma = qq'/N$.[26]

For simplicity, in the following discussion we will assume

$$\eta_k \nu_k = 0 \in H^4(G, \mathbb{Z}_N) \, , \tag{4.8}$$

and thus there is no anomaly.

**Gauge transformation**  The gauge transformation is the following. The group cocycle $\eta_k$ satisfies

$$\eta_k(g^{-1}ag + g^{-1}dg) - \eta_k(a) = d\zeta_k(a, g) \, , \tag{4.9}$$

and similarly for $\nu_k$ with $\xi_k$ replacing $\zeta_k$. The gauge transformation is

$$\begin{aligned} A^k &\to A^k + d\lambda_0^k + \alpha^k - \zeta_k(a, g) \\ B^k &\to B^k + d\lambda_1^k - \xi_k(a, g)\beta^k \\ a &\to g^{-1}ag + g^{-1}dg \, , \end{aligned} \tag{4.10}$$

where $\alpha^k e^k = 0$. We omit an anomalous transformation of $a$, which can be cancelled if the anomaly free condition is imposed.

**Observables**  If $G$ is a global symmetry, then the gauge invariant operators are

$$U_k = e^{i \oint A^k}, \quad V_k = e^{i \oint B^k} \, , \tag{4.11}$$

$U_k$ is a planon in order to be invariant under $\alpha^k$, and it carries a projective representation under $G$. $V_k$ does not have a constraint, although it vanishes when it is supported only on a leaf of foliation $k$.

If $G$ is a gauge group, then on each leaf of foliation $k$ there is gauge field with gauge group $H_k$ given by the extension of $G$ by $\mathbb{Z}_N$, specified by $\eta_k$,

$$1 \to \mathbb{Z}_N \to H_k \to G \to 1 \, . \tag{4.12}$$

The gauge field has action $\omega' \in H^4(H_k, U(1))$ given by the pullback from $\omega \in H^4(G, U(1))$ by the map $H_k \to G$. The operator $U_k$ is decorated by a projective representation of $G$ to form a representation of $H_k$, and in general it obeys non-Abelian fusion, which results in non-Abelian planon. By taking a combination of $U^k$ we can obtain non-Abelian lineons and fractons.

## 4.1  Example: $G = \mathbb{Z}_N$

Consider for instance $G = \mathbb{Z}_N$, and $\eta_k(a) = da/N$, $\nu_k = 0$:

$$\frac{N}{2\pi} \sum_k dA^k B^k + \frac{1}{2\pi} \sum_k daB^k + \frac{N}{2\pi} dab \, , \tag{4.13}$$

where in the last term we include a Lagrangian multiplier $b$, which enforces $a$ to be a $\mathbb{Z}_N$ gauge field. The equations of motion are $NdA^k + da = 0$, $NdB^k = 0$, $\sum dB^k + Ndb = 0$, $Nda = 0$.

---

[26]A similar discussion for Hall conductance is in Appendix E of [58]

The gauge transformations are

$$
\begin{aligned}
A^k &\to A^k + d\lambda_0^k + \alpha^k \\
B^k &\to B^k + d\lambda_1^k \\
a &\to a + d\lambda_0 \\
b &\to b + d\lambda_1 \, .
\end{aligned}
\tag{4.14}
$$

A lattice model is derived in Appendix C.

The theory has a gauge invariant operator $e^{i\int A^k}$ that lives on a leaf of foliation $k$ so that it is invariant under $A^k \to A^k + \alpha^k$. It describes planons. The equation of motion implies that if we take $N$ powers of the planon $e^{i\oint A^k}$, we will find a fully mobile particle $e^{i\int a}$. Similarly, if we take $N$ power of $e^{i\int b}$, we will find the sum $e^{i\int \sum B^k}$. Explicitly,

$$
e^{Ni\int b_{it}dx^i dt + b_{ij}dx^i dx^j} = e^{i\int \sum_k B_{kt}^k dx^k dt + B_{ki}^k dx^k dx^i} \, ,
\tag{4.15}
$$

where if the surface is on $t, z$ plane, then the excitation is along $z$ direction, whose intersection with $x, y$ plane can move on the plane by $B_{zx}^3, B_{zy}^3$ and therefore describes a planon.

Let us compute the correlation function on $\mathbb{R}^4$,[27]

$$
\langle e^{i\oint_\Sigma b} e^{i\oint_{\gamma_k} A^k} \rangle \, .
\tag{4.16}
$$

The operator insertion can be expressed using Poincaré duality as

$$
\int b\delta(\Sigma)^\perp + A^k \delta(\gamma_k)^\perp \, ,
\tag{4.17}
$$

where $\delta(\Sigma)^\perp$ is the Poincaré dual of $\Sigma$, given by a delta function two-form that restricts the integration over spacetime to the surface $\Sigma$, and similarly $\delta(\gamma_k)^\perp$ is a delta function three-form. Then integrating out $A^k$ and $b$ gives

$$
dB^k = -\frac{2\pi}{N}\delta(\gamma_k)^\perp, \quad da = -\frac{2\pi}{N}\delta(\Sigma)^\perp \, ,
\tag{4.18}
$$

where the first equation can be solved on $\mathbb{R}^4$ as $B^k = -\frac{2\pi}{N}\delta(\Sigma_k)^\perp$ with $\gamma_k = \partial\Sigma_k$, and we used $d\delta(\Sigma_k)^\perp = \delta(\gamma_k)^\perp$. The remaining action evaluates to

$$
\frac{1}{2\pi}\int da B^k = \frac{2\pi}{N^2}\int \delta(\Sigma)^\perp \delta(\Sigma_k)^\perp = \frac{2\pi}{N^2}\mathrm{Link}(\Sigma, \gamma_k) \, ,
\tag{4.19}
$$

where Link denotes the linking number. Thus the correlation function is

$$
\langle e^{i\oint_\Sigma b} e^{i\oint_{\gamma_k} A^k} \rangle = e^{\frac{2\pi i}{N^2}\mathrm{Link}(\Sigma, \gamma_k)} \, .
\tag{4.20}
$$

In particular, this implies that $e^{iN\oint b}$ and $e^{iN\oint A^k}$ are non-trivial and satisfy the $\mathbb{Z}_N$ valued correlation function

$$
\langle e^{Ni\oint_\Sigma b} e^{i\oint_{\gamma_k} A^k} \rangle = \langle e^{i\oint_\Sigma b} e^{iN\oint_{\gamma_k} A^k} \rangle = e^{\frac{2\pi i}{N}\mathrm{Link}(\Sigma, \gamma_k)} \, .
\tag{4.21}
$$

In the case $N = 2$, the theory describes the low energy theory of the hybrid toric code layer model in [31]. We will later see that the hybrid toric code model is also described by our magnetic theory (5.22), which we show is dual to this theory in Section 7.1.

---

[27]The computation is similar to that in [47] for non-foliated gauge fields.

We remark that the theory with $A$ and $B$ replaced by ordinary one-form and two-form gauge fields is equivalent to ordinary $\mathbb{Z}_{N^2}$ gauge theory. To see this, note that integrating out $A, b$ allows us to rewrite the action in the discrete notation, with $\overline{B}, \overline{a}$ being $\mathbb{Z}_N$ two-cocycle and one-cocycle: (roughly, $\overline{B} = \frac{N}{2\pi} B^k$ and $\overline{a} = \frac{N}{2\pi} a$)

$$\frac{2\pi}{N} \int \overline{B} \cup \text{Bock}(\overline{a}) \,, \tag{4.22}$$

where Bock is the Bockstein homomorphism for the short exact sequence $1 \to \mathbb{Z}_N \to \mathbb{Z}_{N^2} \to \mathbb{Z}_N \to 1$. Then integrating out $\overline{B}$ enforces $\text{Bock}(\overline{a})$ to be trivial, which implies that $\overline{a}$ can be lifted to a $\mathbb{Z}_{N^2}$ one-cocycle, and the theory describes an ordinary $\mathbb{Z}_{N^2}$ gauge theory. In comparison, here with $A$ and $B$ replaced by constrained fields, the extra particle in the $\mathbb{Z}_{N^2}$ gauge theory is no longer fully mobile but instead constrained to move along a plane.

## 4.2 Example: $G = \mathbb{Z}_2 \times \mathbb{Z}_2$

Consider $G = \mathbb{Z}_2 \times \mathbb{Z}_2$ and $N = 2$, and $\eta_k(a_1 a_2) = a_1 a_2$, $\nu_k = 0$. Then $H_k = \mathbb{D}_8$. The action is given by

$$\frac{2}{2\pi} \sum_k dA^k B^k + \frac{2}{2\pi^2} \sum_k a_1 a_2 B^k + \frac{2}{2\pi} da_1 b_1 + \frac{2}{2\pi} da_2 b_2 \,. \tag{4.23}$$

The equation of motion gives $2dB^k = 0$, $2dA^k = \frac{2}{\pi} a_1 a_2$, $2db_1 = -\frac{2}{\pi} \sum_k a_2 B^k$, $2db_2 = \frac{2}{\pi} \sum_k a_1 B^k$, $2da_1 = 0$, $2da_2 = 0$.

The gauge transformations are

$$A^k \to A^k + d\lambda_0^k + \alpha^k + \frac{1}{\pi} a_1 \lambda_0' - \frac{1}{\pi} \lambda_0 a_2$$

$$B^k \to B^k + d\lambda_1^k$$

$$a_1 \to a_1 + d\lambda_0$$

$$a_2 \to a_2 + d\lambda_0'$$

$$b_1 \to b_1 + d\lambda_1 + \frac{1}{\pi} \sum_k a_2 \lambda_1^k - \frac{1}{\pi} \lambda_0' \sum_k B^k$$

$$b_2 \to b_2 + d\lambda_1' - \frac{1}{\pi} \sum_k a_1 \lambda_1^k + \frac{1}{\pi} \lambda_0 \sum_k B^k \,. \tag{4.24}$$

The operator $e^{i \oint A^k + \frac{1}{\pi} \int a_1 a_2}$ describes the Wilson line in the two-dimensional representation of $H_k = \mathbb{D}_8$, and it transforms under the center. The Wilson lines $e^{i \oint a_1}, e^{i \oint a_2}$, and $e^{i \oint a_1 + a_2}$ are the sign representations of $\mathbb{D}_8$, and they describe fully mobile particles. Thus two fractons can fuse into multiple fully mobile particles, following from the fusion rule of $\mathbb{D}_8$ representations.[28] The invariance under $A^k \to A^k + \alpha^k$ with $\alpha^k e^k = 0$ implies $\oint A^k$ is a planon. Taking nonzero linear combination of $\oint A^{k_1}, \oint A^{k_2}$ then produces a lineon $\oint A^1 - A^2$, which obeys Abelian fusion since $\eta_k$ has order 2. Taking a nonzero linear combination of $\oint A^{k_1}, \oint A^{k_2}, \oint A^{k_3}$ then produces a fracton, which obeys non-Abelian fusion.

## 5 Magnetic model

Consider $G$ gauge theory for some finite or continuous group $G$. We can express the $G$ gauge field as a $G/\mathcal{A}$ gauge field extended by some finite Abelian normal subgroup $\mathcal{A} \subset G$ gauge

---

[28]Another way to see this is that fusing $e^{i \oint A^k + \frac{1}{\pi} \int a_1 a_2} \; e^{i \oint A^k + \frac{1}{\pi} \int (a_1 + d\phi_1)(a_2 + d\phi_2)}$ can produce $e^{i \oint a_1}, e^{i \oint a_2}, e^{i \oint a_1 + a_2}$ depending on the gauge parameters $\phi_1, \phi_2 = 0, \pi$.

field using the group extension:

$$1 \to \mathcal{A} \to G \to G/\mathcal{A} \to 1 \, . \tag{5.1}$$

The $G$ gauge field can be expressed by the pair $(a, a')$ with $a$ being the $\mathcal{A}$ gauge field and $a'$ being the $G/\mathcal{A}$ gauge field, which are constrained to satisfy $da = \eta(a')$ for $\eta \in H^2(G/\mathcal{A}, \mathcal{A})$ specifying the extension $G$. In general, there can be a non-trivial permutation action of $G/\mathcal{A}$ on $\mathcal{A}$ in the group extension, and the cohomology group is understood as generally a twisted cohomology group. In the following we will still use $d$ to denote the corresponding twisted coboundary operator. Since $\mathcal{A}$ is a product of finite cyclic groups, without loss of generality suppose $\mathcal{A} = \mathbb{Z}_N$. Then the condition on the gauge fields can be imposed by a Lagrangian multiplier $b$:

$$\frac{N}{2\pi}(da - \eta(a'))b \, . \tag{5.2}$$

A general Wilson line of $G$ is described by a Wilson line of $\mathcal{A}$ and a projective representation of $G/\mathcal{A}$ from the Mackey theory.

Next, we couple the theory to $(A^k, B^k)$ gauge fields as

$$S_M = \frac{N}{2\pi}\sum_k dA^k B^k + \sum_k \frac{Nq^k}{2\pi}bB^k + \frac{N}{2\pi}(da - \eta(a'))b + S_{\text{top}}(a', B^k) \, , \tag{5.3}$$

where we include a "symmetry twist"

$$S_{\text{top}}(a', B^k) = \omega(a') + \sum_{kl} \frac{Np_{kl}}{4\pi}B^k B^l \, , \tag{5.4}$$

with $\omega \in H^4(G/\mathcal{A}, U(1))$, and integer $p_{kl}$. We call $S_M$ the magnetic model.

Integrating out $b$ now imposes

$$da = \eta(a') - \sum_k q^k B^k \, . \tag{5.5}$$

**Gauge transformation**    Denote

$$\eta(g^{-1}a'g + g^{-1}dg) - \eta(a') = d\zeta(a', g) \, . \tag{5.6}$$

The gauge transformation is

$$\begin{aligned}
A^k &\to A^k + d\lambda_0^k + \alpha^k - q^k\lambda_1 - \sum_l p_{kl}\lambda_1^l \\
B^k &\to B^k + d\lambda_1^k \\
a &\to a + d\lambda_0 + \zeta(a', g) - \sum_k q^k\lambda_1^k, \quad a' \to g^{-1}a'g + g^{-1}dg \\
b &\to b + d\lambda_1 \, .
\end{aligned} \tag{5.7}$$

**Observables**    For simplicity, we will assume $\mathcal{A}$ is in the center of $G$. From (5.5) and (5.7), the $G$ Wilson line that transforms under the center (projective representation of $G/\mathcal{A}$) transforms by $q^k\lambda_1^k$. Thus the gauge invariant operators include the fracton (for all three $q^k$ nonzero)

$$e^{i\oint a} \, , \tag{5.8}$$

which is generally non-Abelian for non-Abelian $G$, as well as a lineon such as

$$e^{i\oint A^1 - A^2 + i\int \sum_l (p_{1l} - p_{2l})B^l} \, , \tag{5.9}$$

which creates a deconfined lineon if $p_{1l} = p_{2l}$, otherwise we would need to take a power of the above operator.

## 5.1 Example: $\mathbb{Z}_N$ X-cube model

When $G = \mathcal{A} = \mathbb{Z}_N$, $q^k = 1$, $\omega = 0, p_{kl} = 0$, the magnetic model describes the foliated QFT of the $\mathbb{Z}_N$ X-cube model in [19],[29]

$$\frac{N}{2\pi} \sum_k dA^k B^k + \frac{N}{2\pi} \sum_k bB^k + \frac{N}{2\pi} bda . \tag{5.10}$$

The theory is studied in [19].

The magnetic model for general group (5.3) with $p_{kl} = 0$ is equivalent to coupling $G = G/\mathbb{Z}_N$ gauge theory, with gauge fields $a'$, to the theory for the X-cube model, with fields $A^k, B^k, a, b$.

$$S_M = S_{G'}(a') - \frac{N}{2\pi} \eta(a')b + S_X(A^k, B^k, a, b), \quad S_{G'}(a') = \omega(a')$$

$$S_X(A^k, B^k, a, b) = \frac{N}{2\pi} \sum_k dA^k B^k + \sum_k \frac{Nq^k}{2\pi} bB^k + \frac{N}{2\pi} dab . \tag{5.11}$$

### 5.1.1 Singularity structure in "bulk field"

Let us use this example to illustrate that the singularity of the original "bulk fields" $a$ and $b$ can change due to coupling to the foliated gauge fields $(A^k, B^k)$.[30]

The equation of motion for $b$ implies $da + B^k = 0$. Since $B^k$ can have singularity $\delta(\mathcal{L}_k)^\perp$ for some leaf $\mathcal{L}_k$ of foliation $k$, the same holds for $da$ after coupling to $A^k, B^k$. After coupling to the foliated gauge field, there these "bulk fields" now develop singularities on the leaves are no longer the original bulk bundle. In other words, the bulk bundle changes into a new bundle with a singularity structure specified by the coupling to the foliated gauge fields $(A^k, B^k)$, and there is no longer a well-defined bulk field without the novel singularity structure.

We remark that this is similar to $SU(2)$ gauge field, when coupled to two-form gauge field by the center one-form symmetry, becomes an $SO(3)$ gauge field, and there is no longer a well-defined $SU(2)$ bundle.

### 5.1.2 Ground State Degeneracy

Let us calculate the ground state degeneracy of the field theory (5.10) which describes the X-cube model. Take the foliation one-forms to be $e^1 = dz$, $e^2 = dy$, and $e^3 = dz$ on a $T^4$ spacetime with lengths $l_0, l_1, l_2, l_3$ in the four directions.[31] Then up to gauge transformations, the equations of motion from integrating out $A_0^k$, $B_{0k}^k$, $a_0$, and $b_{0i}$ for $k = 1, 2, 3$ are

$$\epsilon^{ijl} \partial_i B_{jl}^k = 0, \quad \partial_i A_j^k + b'_{ij} = 0, \quad B_{jk}^k - B_{kj}^j + \partial_j a_k = 0, \quad \epsilon^{0ijk} \partial_i b'_{jk} = 0 . \tag{5.12}$$

---

[29]Here we use a different notation from [19] and call $B^k$ the foliated two-form gauge field $B^k e^k = 0$.

[30]We thank Nathan Seiberg and Shu-Heng Shao for bringing up this point.

[31]See also [17, 22, 32] for other field theory calculations of the X-cube degeneracy.

They can be solved by

$$
\begin{aligned}
A_i^k &= \sum_{j=1,2,3} \epsilon^{ijk} \frac{q_i^k(t,x^k)}{l_i} \\
B_{ik}^k &= \frac{1}{2} \frac{p_i^k(t,x^k) + p_k^i(t,x^i)}{l_i l_k} \\
a_i &= 0 \\
b_{ij}' &= 0
\end{aligned}
\tag{5.13}
$$

$$
\int dx^k q_i^k(t,x^k) = \int dx^k p_i^k(t,x^k) = 0 \text{ when } k-i \equiv 1 \bmod 3 \,,
\tag{5.14}
$$

where $i,j = 1,2,3$.

Plugging the above solution into the action yields

$$
S = \frac{N}{2\pi} \sum_{j,k=1,2,3,j\neq k} \int_0^{l_0} dt \int_0^{l_k} \frac{dx^k}{l_k} \, p_j^k(t,x^k)\partial_0 q_j^k(t,x^k).
\tag{5.15}
$$

The partition function and ground state degeneracy of the above action is divergent. In order to obtain a finite result, we can impose a lattice regularizations in the $x^k$ direction for each foliation with a lattice spacing $\Lambda_k = l_k/L_k$, where $L_k$ is the lattice length in the $k$ direction. This amounts to substituting

$$
p_i^k(t,x^k) = \sum_r \delta(x^k - r\Lambda_k) p_{i,r}^k(t) \,.
\tag{5.16}
$$

We also denote $q_{j,r}^k(t) = q_j^k(t,r\Lambda_k)$. Then the integral over $x^k$ is replaced by a sum in the effective action:

$$
S = \frac{N}{2\pi} \sum_{j,k=1,2,3:j\neq k} \int_0^{l_0} dt \sum_{r=0,1,\ldots,(L_k-1)} p_{j,r}^k(t)\partial_0 q_{j,r}^k(t) \,.
\tag{5.17}
$$

This theory has a ground state degeneracy equal to $N^{2L_1+2L_2+2L_3-3}$.

### 5.1.3 Comparison with twisted foliated two-form gauge theory

Let us compare the set of operators for the $N = 2$ theory of (5.10) describing the $\mathbb{Z}_2$ X-cube model with the twisted $\mathbb{Z}_2$ foliated two-form gauge theory (3.1) for $N = 2, p_{kl} = 1$. We will denote the fields in (5.10) that describes the X-cube model with a tilde:

$$
\begin{aligned}
\text{Fracton} \quad & e^{i\oint A^1+A^2+A^3} \quad \text{v.s.} \quad e^{i\oint \tilde{a}} \\
\text{Fracton-lineon} \quad & e^{i\oint A^1} \quad \text{v.s.} \quad e^{i\oint \tilde{a}+\tilde{A}^2-\tilde{A}^3} \\
\text{Lineon} \quad & e^{i\oint A^1-A^2} \quad \text{v.s.} \quad e^{\oint \tilde{A}^1-\tilde{A}^2} \\
\text{Fracton dipole} \quad & e^{\int B^k} \quad \text{v.s.} \quad e^{i\int \tilde{B}^k} \,,
\end{aligned}
\tag{5.18}
$$

where in the last line the field is integrated over a ribbon whose boundary is the worldline of a pair of fractons. Each pair satisfies the same fusion algebra (up to trivial surfaces such as $2\int B^k, 2\int \tilde{B}^k, 2\int \tilde{b}$ that we ignore here). This is the field theory counterpart for the mapping of fusion module in (3.27). However, the pairs have different statistics, and thus the two theories are not equivalent.

## 5.2 Example: $G = \mathbb{Z}_{N^2}$

Consider $\mathbb{Z}_{N^2}$ described by the extension of $G/\mathcal{A} = \mathbb{Z}_N$ by $\mathbb{Z}_N$. Denote the gauge field of $\mathcal{A}$ by $a$, and $G/\mathcal{A}$ by $a'$ as before. For $S_{\text{top}} = 0$, the magnetic model is

$$S_M = \frac{N}{2\pi} \sum_k dA^k B^k + \frac{N}{2\pi} \sum_k q_k b B^k + \frac{N}{2\pi} dab - \frac{1}{2\pi} da'b + \frac{N}{2\pi} da'b' , \tag{5.19}$$

where we include a Lagrangian multiplier $b'$ to enforce $a'$ to be $\mathbb{Z}_N$ valued. The equation of motion gives $NdB^k = 0$, $NdA^k + q_k Nb = 0$, $Nq_k B^k + Nda - da' = 0$, $-db + Ndb' = 0$, $Nda' = 0$.

We can also integrate out $a'$ as a Lagrangian multiplier, which imposes $b = Nb'$ up to a gauge transformation, and we find the following equivalent theory

$$S'_M = \frac{N}{2\pi} \sum_k dA^k B^k + \sum_k \frac{N^2 q_k}{2\pi} b'B^k + \frac{N^2}{2\pi} dab' , \tag{5.20}$$

with the gauge transformation

$$A^k \rightarrow A^k + d\lambda_0^k + \alpha^k - Nq_k \lambda_1'$$
$$B^k \rightarrow B^k + d\lambda_1^k$$
$$a \rightarrow a + d\lambda_0 - \sum_k q_k \lambda_1^k$$
$$b' \rightarrow b' + d\lambda_1' . \tag{5.21}$$

The equations of motion are $NdB^k = 0$, $NdA + N^2 q_k b' = 0$, $N^2 db' = 0$, $N^2 da + N^2 q_k B^k = 0$. This theory is a special case of (3) in [19] with the substitutions $M_k \rightarrow N$, $N \rightarrow N^2$, and $n_k \rightarrow Nq_k$. A string-membrane-net lattice model for (5.20) is given in Appendix A of [32] with similar substitutions.

Let us study the examples $q_k = (0, 0, 1)$, $q_k = (0, 1, 1)$ and $q_k = (1, 1, 1)$, where we consider a single foliation, two foliations and three foliations for the foliations $k$ with nonzero $q_k$.

**Example 1: single foliation** Let us omit $A^1, B^1, A^2, B^2$. The theory is

$$\frac{N}{2\pi} dA^3 B^3 + \frac{N^2}{2\pi} B^3 b' + \frac{N^2}{2\pi} b' da . \tag{5.22}$$

The theory has a planon particle $e$ described by $e^{i \oint a}$ and a loop excitation $m$ described by $e^{i \oint b'}$. The $N$th power of the planon, $e^N$, is described by $e^{Ni \oint a}$, which can be made gauge invariant by

$$e^{Ni \oint a + Ni \int B^3} . \tag{5.23}$$

Since $N \int B^3$ is trivial, this is a genuine line operator that creates a fully mobile particle. The $N$th power of the loop excitation $e^{Ni \oint b'}$ can end

$$e^{i \oint A^3 + Ni \int b'} , \tag{5.24}$$

where the ending $\oint A^3$ lives on a leaf to be invariant under the gauge transformation $A^3 \rightarrow A^3 + \alpha^3$. If we take $e^3 = dz$, this means it lives on the $x, y, t$ space. Thus the operator describes a planon, denoted by $m^N$, that moves on a leaf.

The ground state degeneracy on a three-torus can be computed similarly to Section 5.1.2, and it can also be computed from a string-membrane-net lattice model for the foliated gauge theory in [32] with the method of [60, 61]. For a spacial three-torus with length $L_z$ in the $z$ direction measured in some lattice cutoff unit, the ground state degeneracy equals $N^{2L_z + 3}$.

For $N = 2$, the theory describes the low energy theory of the hybrid toric code layer model in [31], and we find that the ground state degeneracy of the two theories agree.

**Example 2: two foliations**    We will omit $A^3, B^3$. The theory is

$$\frac{N}{2\pi} dA^1 B^1 + dA^2 B^2 + \frac{N^2}{2\pi}(B^1 + B^2)b' + \frac{N^2}{2\pi} b' da \ . \tag{5.25}$$

The theory has a lineon, denoted by $e$ and described by $e^{i \oint a}$, which moves along the intersection of two leaves of foliation 1 and foliation 2. The theory also has a loop excitation $m$ described by $e^{i \oint b'}$. The theory has another lineon $L$ described by $e^{i \oint A^1 - A^2}$.

The $N$th power of $e$ (i.e. $e^N$) is described by $e^{iN \oint a}$ and is fully mobile since the operator

$$e^{iN \oint a + iN \int (B^1 + B^2)} \tag{5.26}$$

can be defined on any curve, where the surface operator $N \int B^1 + B^2$ is trivial. Thus it is a genuine line operator that describes a deconfined fully mobile particle.

The $N$th power of the loop, described by $e^{iN \oint b'}$, can end on a leaf of either foliation using the gauge invariant operator

$$e^{i \oint A^1 + Ni \int b'}, \quad e^{i \oint A^2 + Ni \int b'} \ . \tag{5.27}$$

In general, the boundary of the surface $e^{iN \int b}$ can be patches that are locally on leaf of either foliation, and at the intersection there is lineon $e^{i \oint A^1 - A^2}$. In other words, the $N$th power of loop $m$ has a lineon $L$ at the corner when turning from the $x$ direction to the $y$ direction. We denote $L = m^N$.

The ground state degeneracy on a three-torus can be computed similarly to Section 5.1.2, and it can also be computed from a string-membrane-net lattice model for the foliated gauge theory in [32] with the method of [60, 61]. For a spatial three-torus with lengths $L_x, L_y$ in the $x, y$ directions measured in some lattice cutoff unit, the ground state degeneracy equals $N^{2L_x + 2L_y + 1}$.

**Example 3: three foliations**    Consider the theory

$$\sum_k \left( \frac{N}{2\pi} dA^k B^k + \frac{N^2}{2\pi} b' B^k \right) + \frac{N^2}{2\pi} b' da \ . \tag{5.28}$$

The gauge invariant operators are

- The operator $e^{i \oint a}$ describes a fracton.

- The $N$th power of fracton

$$e^{iN \oint a + N \int B^k} \ , \tag{5.29}$$

  is fully mobile, where it describes a deconfined particle since the surface $N \int B^k$ is trivial.[32]

- Lineon described by $e^{i \oint A^k - A^l}$.

- Fully mobile particle described by the operator $e^{i \int b'}$ corresponds to magnetic a flux loop, while from the equation of motion of $B^k$, the $N$th power $e^{iN \int b'} = e^{i \int dA^k}$ can live on open surface with boundary, where the boundary can be piecewise smooth where each segment lies on some leaf of foliation $k$. At the intersection point for different leaves $k, l$ there is lineon $e^{i \oint A^k - A^l}$.

---

[32] We remark that in the model (3) of [19], a similar consideration implies that the $r$th power of the basic fracton is fully mobile, with $r = N / \gcd(n_1, n_2, n_3, N)$. When $r \neq N$ it is a non-trivial fully mobile particle. This corrects an imprecise statement in [19].

Let us study the correlation function of the theory on $\mathbb{R}^4$:

$$\langle e^{i\oint_\Sigma b'} e^{i\oint_\gamma a} \rangle = e^{(-2\pi i/N^2)\text{Link}(\gamma,\Sigma)}, \quad \langle e^{i\oint_{\gamma_k} A^k} e^{i\oint_\Sigma B^k} \rangle = e^{(-2\pi i/N)\text{Link}(\gamma_k,\Sigma)}$$

$$\langle e^{i\oint_{\gamma_k} A^k} e^{i\oint_\Sigma b'} \rangle = 1, \quad \langle e^{i\oint_{\Sigma_k} B^k} e^{i\oint_\Sigma b'} \rangle = 1, \quad \langle e^{i\oint_{\Sigma_k} B^k} e^{i\oint_\gamma a} \rangle = 1$$

$$\langle e^{i\oint_{\gamma_k} A^k} e^{i\oint_\gamma a} \rangle = e^{(2\pi i/N)\#(\Sigma,\Sigma_k)}, \quad \partial\Sigma = \gamma, \partial\Sigma_k = \gamma_k . \tag{5.30}$$

Let us compute the last correlation function in detail. The insertion of operator can be expressed using Poincaré duality as

$$\int A^k \delta(\gamma_k)^\perp + a\delta(\gamma)^\perp . \tag{5.31}$$

Integrating out $a, A^k$ gives

$$dB^k = -\frac{2\pi}{N}\delta(\gamma_k)^\perp, \quad db' = -\frac{2\pi}{N^2}\delta(\gamma)^\perp . \tag{5.32}$$

They can be solved on $\mathbb{R}^4$ by

$$B^k = -\frac{2\pi}{N}\delta(\Sigma_k)^\perp, \quad b' = -\frac{2\pi}{N^2}\delta(\Sigma)^\perp , \tag{5.33}$$

where $\partial\Sigma = \gamma, \partial\Sigma_k = \gamma_k$. Then evaluating the remaining action produces the correlation function

$$\langle e^{i\oint_{\gamma_k} A^k} e^{i\oint_\gamma a} \rangle = e^{(2\pi i/N)\int \delta(\Sigma)^\perp \delta(\Sigma_k)^\perp} = e^{(2\pi i/N)\#(\Sigma,\Sigma_k)} . \tag{5.34}$$

The ground state degeneracy on a three-torus can be computed similarly to Section 5.1.2, and it can also be computed from a string-membrane-net lattice model for the foliated gauge theory in [32] with the method of [60, 61]. For space three-torus with lengths $L_x, L_y$ in the $x, y$ directions measured in some lattice cutoff unit, the ground state degeneracy equals $N^{2L_x+2L_y+2L_z}$.

In the case $N = 2$, the theory describes the low energy theory of the fractonic hybrid X-cube model in [31], and indeed the ground state degeneracy agrees.

## 5.3  Example: $G = \mathbb{Z}_{N^2} \times \mathbb{Z}_N$

Consider the theory

$$\sum_k \left( \frac{N}{2\pi} dA^k B^k \right) + \frac{N}{2\pi} \left( Nb(B^1+B^2) + b'(B^2+B^3) \right) + \frac{N^2}{2\pi} bda + \frac{N}{2\pi} b'da' . \tag{5.35}$$

The equations of motion are $NdB^k = 0, NdA^1+N^2b = 0, NdA^2+N^2b+Nb' = 0, NdA^3+Nb' = 0, N^2(B^1+B^2)+N^2da = 0, N(B^2+B^3)+Nda' = 0, N^2db = 0, Ndb' = 0$. The gauge transformations are

$$A^1 \to A^1 + d\lambda_0^1 + \alpha^1 - N\lambda_1$$
$$A^2 \to A^2 + d\lambda_0^2 + \alpha^2 - N\lambda_1 - \lambda_1'$$
$$A^3 \to A^3 + d\lambda_0^1 + \alpha^3 - \lambda_1'$$
$$B^k \to B^k + d\lambda_1^k$$
$$a \to a + d\lambda_0 - \lambda_1^1 - \lambda_1^2$$
$$a' \to a' + d\lambda_0' - \lambda_1^2 - \lambda_1^3$$
$$b \to b + d\lambda_1$$
$$b' \to b' + d\lambda_1' . \tag{5.36}$$

The theory has a lineon, denoted by $e$, which corresponds to the operators $e^{i\oint a}, e^{i\oint a'}, e^{i\oint a+a'}$. The $N$th power can be made gauge invariant by attaching to a trivial surface operator $N\int(B^1-B^2), N\int(B^2-B^3), N\int(B^1-B^3)$, which makes it a fully mobile. Thus, $e^N$ is a fully mobile particle.

The theory has a loop excitation described by $e^{i\oint b}$ of order $N^2$. The $N$th power can be defined on an open surface

$$e^{i\oint A^1+iN\int b}, \quad e^{i\oint A^2-A^3+iN\int b}. \tag{5.37}$$

In particular, the fracton described by

$$e^{i\oint A^1-A^2+A^3} \tag{5.38}$$

can cut in the middle of the surface $e^{iN\oint b}$. [33]

Take $e^1=dx, e^2=dy, e^3=dz$. The ground state degeneracy on a torus of lengths $L_x, L_y, L_z$ along the three directions measured in some lattice cutoff lengths equals $N^{2L_x+2L_y+2L_z}$. The theory with $N=2$ describes the low energy theory of the lineonic hybrid X-cube model in [31]. Indeed, the ground state degeneracy of the two theories can be found to agree.

### 5.3.1 An exactly solvable lattice model

Let us give an exactly solvable lattice model for the foliated gauge theory (5.35). Integrating out $A^k, a, a'$ gives $\mathbb{Z}_N$ gauge fields $B^k, b'$, and $\mathbb{Z}_{N^2}$ gauge field $b$, with the action $\frac{2\pi}{N}\phi_4$: (where we normalize the gauge fields to be $B^k, b'=0,1,\cdots N-1 \bmod N$ and $b=0,1,\cdots N^2-1 \bmod N^2$)

$$\phi_4(B^k, b, b') = b\cup(B^1+B^2) + b'\cup(B^2+B^3). \tag{5.39}$$

It satisfies

$$\phi_4(d\lambda^k, d\widetilde{\lambda}, d\lambda') = d\phi_3, \quad \phi_3(\lambda^k, \widetilde{\lambda}, \lambda') = \widetilde{\lambda}\cup(d\lambda^1+d\lambda^2) + \lambda'\cup(d\lambda^2+d\lambda^3). \tag{5.40}$$

A Hamiltonian for the SPT phase with shift symmetry of $\lambda^k, \widetilde{\lambda}, \lambda'$ is given by conjugating $H_0 = -\sum_{n=0}^{N^2-1}\sum\widetilde{X}_e^n - \sum_{m=0}^{N-1}\left(\sum X_{e_k}^m + \sum X_e'^m\right)$ by $e^{\frac{2\pi i}{N}\int\phi_3}$:

$$\begin{aligned}
H_{\text{SPT}} = &-\sum_m\sum X_{e_x}^m e^{\frac{2\pi i}{N}\int\phi_3(\lambda^1+m\widetilde{e}_x,\lambda^2,\lambda^3,\widetilde{\lambda},\lambda')-\phi_3(\lambda^1,\lambda^2,\lambda^3,\widetilde{\lambda},\lambda')} \\
&-\sum_m\sum X_{e_y} e^{\frac{2\pi i}{N}\int\phi_3(\lambda^1,\lambda^2+m\widetilde{e}_y,\lambda^3,\widetilde{\lambda},\lambda')-\phi_3(\lambda^1,\lambda^2,\lambda^3,\widetilde{\lambda},\lambda')} \\
&-\sum_m\sum X_{e_z} e^{\frac{2\pi i}{N}\int\phi_3(\lambda^1,\lambda^2,\lambda^3+m\widetilde{e}_z,\widetilde{\lambda},\lambda')-\phi_3(\lambda^1,\lambda^2,\lambda^3,\widetilde{\lambda},\lambda')} \\
&-\sum_n\sum\widetilde{X}_e e^{\frac{2\pi i}{N}\int\phi_3(\lambda^k,\widetilde{\lambda}+n\widetilde{e},\lambda')-\phi_3(\lambda^k,\widetilde{\lambda},\lambda')} - \sum_m\sum X_e' e^{\frac{2\pi i}{N}\int\phi_3(\lambda^k,\widetilde{\lambda},\lambda'+m\widetilde{e})-\phi_3(\lambda^k,\widetilde{\lambda},\lambda')}
\end{aligned} \tag{5.41}$$

Explicitly,

$$\begin{aligned}
H_{\text{SPT}} = &-\sum_{m=0}^{N-1}\left(\sum X_{e_x}^m e^{\frac{2\pi im}{N}\int d\widetilde{\lambda}\cup\widetilde{e}_x} + \sum X_{e_y}^m e^{\frac{2\pi im}{N}\int(d\widetilde{\lambda}+d\lambda')\cup\widetilde{e}_y} + \sum X_{e_z}^m e^{\frac{2\pi im}{N}\int d\lambda'\cup\widetilde{e}_z}\right) \\
&-\sum_{n=0}^{N^2-1}\sum_e\widetilde{X}_e^n e^{\frac{2\pi in}{N}\int\widetilde{e}\cup(d\lambda^1+d\lambda^2)} - \sum_{m=0}^{N-1}\sum_e X_e'^m e^{\frac{2\pi im}{N}\int\widetilde{e}\cup(d\lambda^2+d\lambda^3)}.
\end{aligned} \tag{5.42}$$

---

[33]There is also operator $e^{i\oint b'}$ that can be defined on open surface $e^{i\oint A^3+i\int b'}$, so it does not correspond to non-trivial loop.

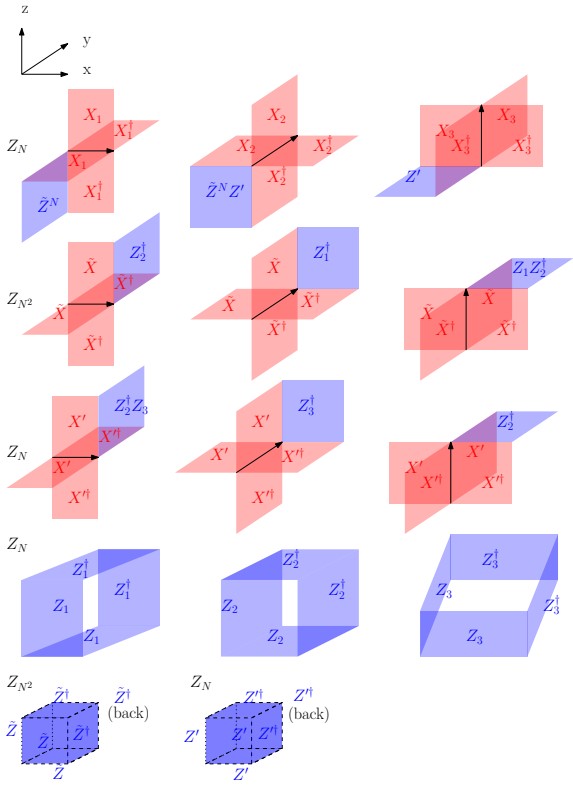

Figure 7: Hamiltonian model for the $\mathbb{Z}_N \times \mathbb{Z}_{N^2}$ magnetic model. The row with the $\mathbb{Z}_N$ in the front means summing over $m = 0, 1, \cdots N-1$ powers of such term, and similarly the row with $\mathbb{Z}_{N^2}$ in the front means summing over $n = 0, 1, \cdots N^2-1$ powers of such terms. The variables $\widetilde{X}, \widetilde{Z}$ are the $\mathbb{Z}_{N^2}$ analogue of Pauli $X, Z$ matrices, while the other variables are the $\mathbb{Z}_N$ analogue of Pauli matrices.

After gauging the shift symmetries, the Hamiltonian model is

$$
\begin{aligned}
H_{\text{gauged}} = -\sum_{m=0}^{N-1} &\left( \sum_{e_x} \left(\prod X_{f_x}\right)^m e^{\frac{2\pi i m}{N} \int b \cup \widetilde{e}_x} + \sum_{e_y} \left(\prod X_{f_y}\right)^m e^{\frac{2\pi i m}{N} \int (b+b') \cup \widetilde{e}_y} \right. \\
&\left. + \sum_{e_z} \left(\prod X_{f_z}\right)^m e^{\frac{2\pi i m}{N} \int b' \cup \widetilde{e}_z} \right) \\
&- \sum_{n=0}^{N^2-1} \sum_e \left(\prod \widetilde{X}_f\right)^n \widetilde{e}^{\frac{2\pi i n}{N} \int \widetilde{e} \cup (B^1 + B^2)} - \sum_{m=0}^{N-1} \sum_e \left(\prod X_f'\right)^m e^{\frac{2\pi i m}{N} \int \widetilde{e} \cup (B^2 + B^3)} \\
&- \sum_{c_k} \prod Z_{f_k} - \sum_c \prod \widetilde{Z}_f - \sum_c \prod Z_f' .
\end{aligned}
\tag{5.43}
$$

The faces on $x^i - x^j$ plane has $\mathbb{Z}_N$ degrees of freedom associated with $B^i, B^j, b'$ and $\mathbb{Z}_{N^2}$ degrees of freedom associated with $b'$. The terms in the Hamiltonian model are shown in Figure 7.

The ground state degeneracy on a torus of lengths $L_x, L_y, L_z$ along the three directions measured in the unit of lattice spacing can be computed using the method of [60, 61] and it equals $N^{2L_x + 2L_y + 2L_z}$.

## 5.4 Example: $G = \mathbb{D}_8$

Consider $G$ as the extension of $\mathbb{Z}_2 \times \mathbb{Z}_2$ by $\mathcal{A} = \mathbb{Z}_2$

$$1 \to \mathbb{Z}_2 \to G \to \mathbb{Z}_2 \times \mathbb{Z}_2 \to 1 \ . \tag{5.44}$$

The extension corresponds to $\eta(a') = a_1 a_2$ for $\mathbb{Z}_2 \times \mathbb{Z}_2$ gauge field $a_1, a_2$. The magnetic model is described by the action

$$\sum_k \frac{2}{2\pi} dA^k B^k + \sum_k \frac{2}{2\pi} b B^k + \frac{2}{2\pi}(da - \frac{1}{\pi}a_1 a_2)b + \frac{2}{2\pi} da_1 b_1 + \frac{2}{2\pi} da_2 b_2 \ , \tag{5.45}$$

where we include Lagrangian multipliers $b_1$ and $b_2$ to enforce $a_1$ and $a_2$ to be $\mathbb{Z}_2$ gauge fields. The equation of motion for $b$ gives

$$\frac{2}{2\pi}\sum_k B^k + \frac{2}{2\pi}(da - \frac{1}{\pi}a_1 a_2) = 0 \ . \tag{5.46}$$

Thus under the gauge transformation: $B^k \to B^k + d\lambda_B^k, \ a \to a - \lambda^k$.[34] The theory has a non-Abelian fracton, described by

$$e^{i\oint_\gamma a - \frac{i}{\pi}\int_\Sigma a_1 a_2} \ , \tag{5.48}$$

where $\gamma$ is the boundary of $\Sigma$, and it lies on the intersection of the leaves for all three foliations. It is the Wilson line in the two-dimensional representation of $\mathbb{D}_8$. On the other hand, the Wilson lines $e^{i\oint a_1}, e^{i\oint a_2}$ and $e^{i\oint a_1 + a_2}$ are the sign representations of $\mathbb{D}_8$, and they are fully mobile. Thus two fractons can fuse into multiple fully mobile particles. The theory also has lineons, described by $e^{i\oint A^k - A^l}$ for $k \neq l$.[35]

The theory has the same spectrum as the electric model discussed in Section 4.2. In fact, as we will discuss in Section 7, the electric and magnetic models are in fact dual to each other.

The theory also appears to have the same kind of planon, lineon, and non-abelian fracton excitations (and no abelian fractons) as the non-abelian $D_4$ model in [66] (see also [67]). ($D_4$ in [66] and $\mathbb{D}_8$ in our work both denote the dihedral group with 8 elements.) For example, both theories have $Z_2$ gauge theory planons and lineons that have a $\pi$ statistic with the fracton. We therefore conjecture that the two models describe the same physics.

We can also replace $\mathcal{A} = \mathbb{Z}_2$ by other Abelian normal subgroups in $\mathbb{D}_8$, which produce different theories. For each normal subgroup $\mathcal{A}$, we can also include topological action for the $G/\mathcal{A}$ gauge field. The discussion is straightforward, and we do not work out the details here.

---

[34]The other gauge transformations are

$$A^k \to A^k + d\lambda_0^k + \alpha^k - \lambda_b$$
$$b \to b + d\lambda_b$$
$$B^k \to B^k + d\lambda_B^k + \frac{1}{\pi}\delta_{k,1}(d\lambda_{a_1}a_2 + a_1 d\lambda_{a_2})$$
$$a \to a - \sum_k \lambda_B^k + \frac{1}{\pi}\lambda_{a_1}d\lambda_{a_2}$$
$$b_1 \to b_1 + d\lambda_{b_1} - \frac{1}{\pi}a_2\lambda_b, \quad b_2 \to b_2 + d\lambda_{b_2} + \frac{1}{\pi}a_1\lambda_b$$
$$a_1 \to a_1 + d\lambda_{a_1}, \quad a_2 \to a_2 + d\lambda_{a_2} \ . \tag{5.47}$$

[35]The lineon has $\pi$ statistics with the fracton: consider the correlation function of $\oint_{\gamma_{12}} A^1 - A^2$ and $\oint_\gamma a - \frac{1}{\pi}\int_\Sigma a_1 a_2$, integrating out the fields gives the correlation function

$$\exp \pi i \int (\delta(\Sigma_{12})^\perp - \delta(\widetilde{\Sigma}_{12})^\perp)\delta(\Sigma)^\perp \ , \tag{5.49}$$

where $\partial\Sigma_{12} = \partial\widetilde{\Sigma}_{12} = \gamma_{12}$.

# 6 Couple foliated gauge field to matter

## 6.1 Couple matter to foliated gauge field by one-form symmetry

Consider a theory with $\mathbb{Z}_N$ one-form symmetry, with background denoted by $B$. We can couple the theory to a $\mathbb{Z}_N$ theory $(A^k, B^k)$ by

$$B = \sum_k c_k B^k \,, \tag{6.1}$$

where $c_k$ are integers mod $N$. This is consistent since the theory can be coupled to a general $\mathbb{Z}_N$ two-form gauge field $B$, and thus can be coupled to $B^k$, which is a special two-form gauge field.

In the electric model $S_E$, the one-form symmetry is generated by $e^{\frac{2\pi}{N}\int \eta_k(a)}$, which we coupled to the gauge field $B^k$. In the magnetic model $S_M$ for central extension, the one-form symmetry is the center one-form symmetry of $G$ gauge theory.

**Example:** $O(3)$ **sigma model**  Consider a sigma model with target space $\mathcal{M}$. The sigma model can have strings, given by $\pi_2(\mathcal{M})$, which describe the one-form symmetry of the model.[36] For instance, all $\mathbb{CP}^{n-1}$ models have $\mathbb{Z}$ strings, *i.e.* $U(1)$ one-form symmetry. We can then couple the model to $(A^k, B^k)$. Consider the simplest case $n = 2$, $\mathcal{M} = S^2$. It is described by a unit vector $n(x) = (n_1, n_2, n_3) \in \mathbb{R}^3$, $n_I n^I = 1$. Denote the skyrmion density by $Q[n]$, which is an integral class of degree two,

$$Q[n] = \frac{1}{8\pi} n_1 \cdot dn_2 \times dn_3 \,. \tag{6.2}$$

The operator $\exp(i\alpha \oint Q[n])$ generates the $U(1)$ one-form symmetry, with parameter $\alpha \in \mathbb{R}/2\pi\mathbb{Z}$. Then one can couple the theory to $(A^k, B^k)$ as

$$S = \frac{N}{2\pi} \sum_k dA^k B^k + \sum_k \frac{Nq^k}{2\pi} B^k Q[n] + \frac{1}{g}(\partial n)^2 \,. \tag{6.3}$$

The gauging transformation $B^k \to B^k + d\lambda^k$ changes the coupling $\frac{Nq^k}{2\pi} B^k Q[n]$ by $\frac{Nq^k}{2\pi} d\lambda^k Q[n]$, which implies that the skyrmions, defined by $\oint Q[n] = 1$ on the transverse surrounding sphere, transforms by $e^{\frac{Nq^k}{2\pi}\oint \lambda^k}$. Thus the gauge invariance implies that skyrmions live on a leaf of foliation $k$ if $q^k \neq 0$, and if $q^k \neq 0$ for all $k$ then the skyrmion in this model becomes a fracton. One the other hand, it can be made gauge invariant by attaching it to $e^{iq^k \int B^k}$, and using the property that $N \int B^k$ is trivial, we conclude that the skyrmion is fully mobile if $Q$ equals a multiple of $N/\gcd(q_1, q_2, q_3, N)$.

**Example:** $SU(N)$ **and** $PSU(N)$ **Yang-Mills theory**  The $SU(N)$ Yang-Mills theory with $\theta = 0$ is believed to have monopole condensation and confinement. It follows that by gauging the center one-form symmetry, $PSU(N)$ gauge theory has deconfined 't Hooft lines, described by a $\mathbb{Z}_2$ two-form gauge theory at low energy (which is also equivalent to $\mathbb{Z}_2$ one-form gauge theory). It has a new magnetic $\mathbb{Z}_N$ one-form symmetry that transforms the 't Hooft lines, and gauging this one-form symmetry with suitable local counterterm recovers the confined $SU(N)$ gauge theory.

Let us replace the two-form gauge field by a foliated two-form gauge field. For instance, we can define a new version of $SU(N)$ gauge theory by coupling the $PSU(N)$ gauge theory to the

---

[36]The non-trivial element of $\pi_2(\mathcal{M})$ gives an operator on $S^2$ whose eigenvalue measures the charge of the lines surround by $S^2$. For general spacetime dimension $D$, the one-form symmetry is described by $\pi_{D-2}(\mathcal{M})$.

foliated two-form gauge theory $\frac{N}{2\pi}\sum_k dA^k B^k$ using the magnetic one-form symmetry, instead of coupling to an ordinary two-form gauge field that would give the ordinary $SU(N)$ gauge theory. At low energy, the theory becomes equivalent to the low energy of the X-cube model, where the 't Hooft line becomes a fracton, and the new Wilson line is a lineon. Thus the theory becomes deconfined, in contrary to the ordinary $SU(N)$ gauge theory where all particles are fully mobile but confined. Similarly, we can consider a new $PSU(N)$ gauge theory by coupling the $SU(N)$ gauge theory to the foliated two-form gauge theory instead of ordinary two-form gauge field, then the 't Hooft line becomes a planon $e^{i\oint A^k}$, while the Wilson line can end on the surface $e^{i\int \Sigma B^k}$ and it is a confined fracton. The discussion can be generalized to include $\theta$ angles.

## 6.2 Couple matter to foliated gauge field by ordinary or planar symmetry

Let us discuss some examples of coupling the foliated gauge field to matter using a symmetry that acts on the leaves of foliation.

### 6.2.1 Example: magnetic model with matter

Let us discuss foliated gauge theory with a minimal coupling to matter fields:

$$
\begin{aligned}
L = \sum_k \frac{M_k}{2\pi}\left(dA^k + n_k b\right)B^k + \frac{N}{2\pi}bda \\
+ g_1^{-1}\sum_k \left|e^k(d\Theta^k - n_k\phi - A^k)\right|^2 + g_2^{-1}\sum_k \left(d\Phi^k - B^k\right)^2 \\
+ g_3^{-1}\sum_k \left(d\theta - \sum_k m_k\Phi^k - a\right)^2 + g_4^{-1}(d\phi - b)^2 .
\end{aligned}
\tag{6.4}
$$

$\Theta^k$ and $\theta$ are 0-form matter fields while $\Phi^k$ and $\phi$ are 1-form matter fields. The last two lines are obtained by considering gauge transformations for each gauge field, and then replacing each gauge parameter with a matter field that transforms the replaced gauge parameter:

$$
\begin{aligned}
A^k &\to A^k + d\lambda_0^k - n_k\lambda_1 + \alpha^k & \Theta^k &\to \Theta^k + \lambda_0^k \\
B^k &\to B^k + d\lambda_1^k & \Phi^k &\to \Phi^k + \lambda_1^k \\
a &\to a + d\lambda_0 - \sum_k m_k\lambda_1^k & \theta &\to \theta + \lambda_0 \\
b &\to b + d\lambda_1 & \phi &\to \phi + \lambda_1 .
\end{aligned}
\tag{6.5}
$$

When any of the $g_i^{-1}$ are sufficiently large, some of the gauge fields will be Higgsed by the matter field.

## 6.3 Faithful symmetry on leaves

Consider independent degrees of freedom living on different foliation, acted by $K \times G$ symmetry. Moreover, the symmetry action on foliation $k$ has $\mathbb{Z}_{N_k}$ identification: the faithful symmetry for foliation $k$ is

$$
\frac{K \times G}{\mathbb{Z}_{N_k}} .
\tag{6.6}
$$

Now, let us gauge the $K$ symmetry, which turns the theory into a $K$ gauge theory. Then the faithful symmetry for foliation $k$ is

$$
G' = G/\mathbb{Z}_{N_k} .
\tag{6.7}
$$

While the $G$ action leaves the theory invariant, it is important to consider the faithful symmetry $G'$. For instance, by turning on a $G'$ background gauge field we can use discrete anomalies to study the theory, such as in [68, 69].

To describe the $G'$ gauge field, we can use the magnetic model. It is a $G$ gauge field and $\mathbb{Z}_{N_k}$ foliated two-form gauge field described by $(A^k, B^k)$. For instance, if $N_k = N$, then the $G'$ gauge field is described by

$$\sum_k \frac{N}{2\pi} dA^k B^k + \frac{N}{2\pi} \sum_k b B^k + \frac{N}{2\pi}(da - \eta(a'))b \, , \tag{6.8}$$

which is the magnetic model for $\mathcal{A} = \mathbb{Z}_N$, and $\eta \in H^2(G/\mathbb{Z}_N, \mathbb{Z}_N)$ is specified by the extension $G$. A $G'$ background gauge field is described by classical fields $B^k, a, a'$ constrained to satisfy

$$da = \eta(a') - \sum_k B^k \, , \tag{6.9}$$

which is enforced by integrating out $A^k, b$.[37]

For instance, consider a $K = U(1)$ foliated gauge field $A$ coupled to $N_f$ scalars of charge one on a leaf. The theory has $G' = PSU(N_f) \times U(1)$ planar symmetry. The $G'$ symmetry has an anomaly from a mixed anomaly between the $SPU(N_f)$ symmetry and $U(1)$ symmetry [68].

# 7 Dualities for foliated gauge theories

## 7.1 Example: $\mathbb{Z}_N$ and $\mathbb{Z}_{N^2}$ model

Let us begin with a simple example of duality between the electric model and magnetic model,

$$
\begin{aligned}
\text{Electric:} \quad & \frac{N}{2\pi} dAB + \frac{1}{2\pi} Bda + \frac{N}{2\pi} bda \\
\text{Magnetic:} \quad & \frac{N}{2\pi} d\widetilde{A}\widetilde{B} + \frac{N^2}{2\pi} \widetilde{B}\widetilde{b} - \frac{N^2}{2\pi} \widetilde{b}d\widetilde{a} \, .
\end{aligned} \tag{7.1}
$$

Here, we only consider a single foliation. However, the duality generalizes naturally to additional foliations.

Let us start with the electric model. First, redefine $a' = a + NA$. It has the gauge transformation

$$
\begin{aligned}
& a \to a + dl, \quad A \to A + d\lambda_0 + \alpha \\
& a' \to a' + d(l + N\lambda_0) + N\alpha \, .
\end{aligned} \tag{7.2}
$$

The action becomes

$$\frac{1}{2\pi} Bda' + \frac{N}{2\pi} bda' - \frac{N^2}{2\pi} bd\widetilde{a} \, , \tag{7.3}$$

where $\widetilde{a} = A$. We can trade $a'$ by the $\mathbb{Z}_N$ two-form

$$\widetilde{B} = \frac{da'}{N}, \quad \widetilde{B} \to \widetilde{B} + d\alpha \, . \tag{7.4}$$

The action can be written as

$$\frac{N}{2\pi} B\widetilde{B} + \frac{N^2}{2\pi} b\widetilde{B} - \frac{N^2}{2\pi} bd\widetilde{a} \, . \tag{7.5}$$

---

[37]If $B^k$ were ordinary two-form gauge field, then $a$ can be removed by a background gauge transformation of $B^k$, and the equation would imply that $\eta(a)$ (the obstruction to lifting the $G/\mathbb{Z}_N$ bundle to a $G$ bundle) equals $\sum B^k$.

So far it is a rewriting using $\widetilde{B}$ satisfying the constraint (7.4). We can relax the constraint by introducing a Lagrangian multiplier $\widetilde{A}$, and integrating out $B$, which imposes $\widetilde{B}e = 0$. We end up with the action

$$\frac{N}{2\pi}d\widetilde{A}\widetilde{B} + \frac{N^2}{2\pi}b\widetilde{B} - \frac{N^2}{2\pi}bd\widetilde{a}, \quad \widetilde{B}e = 0 . \tag{7.6}$$

Thus we recover the magnetic model,

$$\frac{N}{2\pi}dAB + \frac{1}{2\pi}Bda + \frac{N}{2\pi}bda \quad \longleftrightarrow \quad \frac{N}{2\pi}d\widetilde{A}\widetilde{B} + \frac{N^2}{2\pi}\widetilde{B}\widetilde{b} - \frac{N^2}{2\pi}\widetilde{b}d\widetilde{a} . \tag{7.7}$$

The operators map as

$$
\begin{aligned}
e^{i\oint A} \quad &\longleftrightarrow \quad e^{i\oint \widetilde{a}} \\
e^{i\oint a} \quad &\longleftrightarrow \quad e^{iN\oint \widetilde{a} - iN\int \widetilde{B}} \\
e^{i\oint b} \quad &\longleftrightarrow \quad e^{i\oint \widetilde{b}} \\
e^{i\int_\Sigma B} \quad &\longleftrightarrow \quad e^{i\oint_{\partial\Sigma} \widetilde{A} + Ni\oint_\Sigma \widetilde{b}} ,
\end{aligned}
\tag{7.8}
$$

where in the last line $\partial\Sigma$ is on the leaf. One can verify that the corresponding correlation functions agree.

## 7.2 Duality for general group

We will show the duality can be generalized to

$$
\begin{aligned}
\text{Electric:} \quad & \frac{N}{2\pi}dAB - \frac{N}{2\pi}B\eta(a') + S_{\text{top}}(a') + I[a', \phi] \\
\text{Magnetic:} \quad & \frac{N}{2\pi}d\widetilde{A}\widetilde{B} + \frac{N}{2\pi}\widetilde{b}\widetilde{B} + \frac{N}{2\pi}(d\widetilde{a} - \eta(\widetilde{a}'))\widetilde{b} + S_{\text{top}}(\widetilde{a}') + I[\widetilde{a}', \widetilde{\phi}] .
\end{aligned}
\tag{7.9}
$$

For simplicity, we focus on single foliation, and omit the superscript $k$ in the foliated gauge fields $(A^k, B^k), (\widetilde{A}^k, \widetilde{B}^k)$. The gauge groups in the electric and magnetic models that couple to $(A^k, B^k), (\widetilde{A}^k, \widetilde{B}^k)$ are related by $G_{\text{electric}} = G_{\text{magnetic}}/\mathbb{Z}_N$, with gauge field denoted by $a'$ in the electric model and $\widetilde{a}'$ in the magnetic model. In the duality (7.7), $G_{\text{magnetic}} = \mathbb{Z}_{N^2}$ and $G_{\text{electric}} = \mathbb{Z}_N$. In general, the groups can be both non-Abelian, or the electric group $G_{\text{electric}}$ is Abelian and the magnetic group $G_{\text{magnetic}}$ is non-Abelian. In the above duality, $I$ is a coupling of $a'$ to other fields that can be gauge fields or matter fields, collectively denoted by $\phi$, which do not participate in the duality: $a' \leftrightarrow \widetilde{a}'$, $\phi \leftrightarrow \widetilde{\phi}$. For simplicity we will drop the $I$ term in the following derivation of the duality.

### 7.2.1 A derivation of the duality

Let us start with the magnetic model

$$S_M = \frac{N}{2\pi}d\widetilde{A}\widetilde{B} + \frac{N}{2\pi}\widetilde{b}\widetilde{B} + \frac{N}{2\pi}(d\widetilde{a} - \eta(\widetilde{a}'))\widetilde{b} + S_{\text{top}}(\widetilde{a}') . \tag{7.10}$$

The gauge transformation $\widetilde{B} \to \widetilde{B} + d\lambda_1$ also transforms $\widetilde{a} \to \widetilde{a} - \lambda_1$ with $\lambda_1 e = 0$.

Reversing the previous steps, integrating out $\widetilde{A}$ imposes $\widetilde{B} = du'/N$ for some $u' = u - N\widetilde{a}$,[38] and we include another foliated two-form $B$ to impose the condition $\widetilde{B}e = 0$. Denote $A = \widetilde{a}$, $a' = \widetilde{a}'$; the action becomes

$$-\frac{1}{2\pi}B(du - NdA) + \frac{1}{2\pi}\widetilde{b}(du - NdA) + \frac{N}{2\pi}(dA - \eta(a'))\widetilde{b} + S_{\text{top}}(a')$$

$$= \frac{N}{2\pi}BdA - \frac{1}{2\pi}Bdu + \frac{1}{2\pi}\widetilde{b}du - \frac{N}{2\pi}\eta(a')b + S_{\text{top}}(a') . \tag{7.11}$$

---

[38] We include $\widetilde{a}$ such that $u'$ thus defined does not transform under $\lambda_1$.

After integrating out $b$, which imposes $du = N\eta(a')$, we find the dual electric model for $G_{\text{electric}} = G_{\text{magnetic}}/\mathbb{Z}_N$

$$S_E = \frac{N}{2\pi} dAB - \frac{N}{2\pi} B\eta(a') + S_{\text{top}}(a') \, . \tag{7.12}$$

In other words,

$$\frac{N}{2\pi} dAB - \frac{N}{2\pi} B\eta(a') + S_{\text{top}}(a') \quad \longleftrightarrow \quad \frac{N}{2\pi} d\widetilde{A}\widetilde{B} + \frac{N}{2\pi} \widetilde{b}\widetilde{B} + \frac{N}{2\pi} (d\widetilde{a} - \eta(\widetilde{a}'))\widetilde{b} + S_{\text{top}}(\widetilde{a}') \, . \tag{7.13}$$

The example (7.7) corresponds to $G_{\text{magnetic}} = \mathbb{Z}_{N^2}$, which gives $\eta(a') = da'/N$, $S_{\text{top}} = 0$, and include a Lagrangian multipliers $b', \widetilde{b}'$ to enforce $a', \widetilde{a}'$ to be $\mathbb{Z}_N$ gauge fields by $\frac{N}{2\pi} b' da'$ on the left hand side and $\frac{N}{2\pi} \widetilde{b}' d\widetilde{a}'$ on the right hand side, respectively. Then on the right hand side, integrating out $\widetilde{a}'$ then imposes $\widetilde{b} = N\widetilde{b}'$, and this recovers the right hand side of (7.7).

The duality can be interpreted as follows. In the magnetic model, the gauge field $(\widetilde{A}^k, \widetilde{B}^k)$ couples to the center $\mathbb{Z}_N$ one-form symmetry for $G_{\text{magnetic}}$ associated with $G_{\text{magnetic}}/\mathbb{Z}_N = G_{\text{electric}}$. Gauging the one-form symmetry using ordinary $\mathbb{Z}_N$ two-form gauge fields would result in confined Wilson lines transformed under the center; instead, here we use a constrained gauge field, thus the Wilson lines transformed under the center are deconfined but have restricted mobility. These lines are $e^{i\oint \widetilde{a}}$. In the electric model, the gauge group is the quotient $G_{\text{magnetic}}/\mathbb{Z}_N = G_{\text{electric}}$, and gauging the $\mathbb{Z}_N$ one-form symmetry associated with the quotient extends the gauge group to be $G_{\text{magnetic}}$, which introduces new Wilson lines; here, we couple the constrained gauge fields $(A^k, B^k)$ to the one-form symmetry, thus the new Wilson lines have restricted mobility. These lines are $e^{i\oint A}$. The Wilson lines with full mobility are those of $G_{\text{magnetic}}/\mathbb{Z}_N = G_{\text{electric}}$ and are the same in the two models, namely $e^{i\oint a'}$ and $e^{i\oint \widetilde{a}'}$. The "remaining" Wilson lines have restricted mobility in both models, and correspond to each other under the duality map $A \longleftrightarrow \widetilde{a}$.

We remark that if we replace the foliated gauge fields by ordinary one-form and two-form gauge fields, then the duality would not hold: the electric model would be equivalent to a $G_{\text{magnetic}}$ gauge theory, while the magnetic model would be equivalent to $G_{\text{magnetic}}/\mathbb{Z}_N$ gauge theory. They are in general inequivalent.

## 7.3 Duality for $U(1)$ gauge theory

From the above reasoning, we can consider the following duality for foliated $U(1)$ gauge theory. The electric model is given by gauging a $\mathbb{Z}_N$ subgroup magnetic symmetry in the foliated $U(1)/\mathbb{Z}_N$ gauge theory, while the magnetic model is given by gauging a $\mathbb{Z}_N$ subgroup electric symmetry in the foliated $U(1)$ gauge theory. The two theories are dual. In the magnetic model, the basic Wilson line with charge $\widetilde{q}_e \notin N\mathbb{Z}$ becomes a planon, while Wilson line with charge equals a multiple of $N$ is fully mobile. In the electric model, before gauging the symmetry the Wilson lines have charge equals a multiple of $N$, and they remain fully mobile after gauging the symmetry; after gauging the $\mathbb{Z}_N$ magnetic symmetry, there are new Wilson lines with charge $q \notin N\mathbb{Z}$, and they are planons.

We can also consider duality between ordinary $U(1)$ gauge theories with matter coupled to a foliated gauge field. For instance, take the electric model to be quantum electrodynamics (QED) with even number $N_f$ of charge-one fermions coupled to a $\mathbb{Z}_q$ foliated gauge field (described by $\frac{q}{2\pi} dA_1 B_2$) by the $\mathbb{Z}_q \subset U(1)$ subgroup magnetic one-form symmetry [34] $\eta \in H^2(U(1), \mathbb{Z}_q) = \mathbb{Z}_q$. This implies that the Wilson line has $1/q$ charge under $A_1$. The dual magnetic model is given by QED with charge $q$ fermion, coupled to a $\mathbb{Z}_q$ foliated gauge field

by the $\mathbb{Z}_q$ electric one-form symmetry [34]. Denoting the $U(1)$ gauge field by $u$, the duality is

$$\text{Electric model: } U(1)_u \text{ with } N_f \, \psi \text{ of charge } 1 \, + \frac{q}{2\pi} du B + \frac{q}{2\pi} dAB$$

$$\text{Magnetic model: } U(1)_u \text{ with } N_f \, \widetilde{\psi} \text{ of charge } q + \frac{2q}{e^2} \star du\widetilde{B} + \frac{q}{2\pi} d\widetilde{A}\widetilde{B} \, , \qquad (7.14)$$

where $e$ is the gauge coupling.

# Acknowledgement

We thank Xie Chen, Hotat Lam, Nathan Seiberg, Shu-Heng Shao, Wilbur Shirley, and Nathanan Tantivasadakarn for discussions. The work of P.-S. H. is supported by the U.S. Department of Energy, Office of Science, Office of High Energy Physics, under Award Number DE-SC0011632, and by the Simons Foundation through the Simons Investigator Award. K.S. is supported by the Walter Burke Institute for Theoretical Physics at Caltech.

# A    Foliated field with singularity as defect insertion

In this note we discuss fields where the path integral sums over certain type of singularities or discontinuities in the fields. The following is a remark on such fields. We will not make further use of this interpretation in the rest of the note.

The path integral of such fields with singularities or discontinuities can be interpreted in two steps. First, we sum over a field with fixed singularity. This is equivalent to the path integral over a continuous field configuration, but with a fixed defect inserted at the singularity:

$$Z[\Sigma] = \int \mathcal{D}b \, e^{iS[b]} U_\Sigma \, . \qquad (A.1)$$

Next, we sum over all the possible insertions of a certain class of defects; this amounts to summing over fields with this class of singularities. If the defect generates a symmetry, then the defect insertion is equivalent to turning on a background $B = \text{PD}(\Sigma)$ where PD denotes the Poincaré dual, and we will denote $Z[\Sigma] = Z[B]$. Then summing over the defect insertions is equivalent to gauging the symmetry, since additional insertion of the symmetry defect does not change the new path integral:

$$Z = \sum_B Z[B] \, . \qquad (A.2)$$

Let us give an example. Take the original field is a compact scalar $\phi(x, y)$ with $x \sim x + l_x$, $y \sim y + l_y$. It has shift symmetry. Then consider the discontinuous configuration

$$\phi(x, y) = 2\pi h(x - x_0)\frac{y}{l_y} + 2\pi h(y - y_0)\frac{x}{l_x} - 2\pi \frac{xy}{l_x l_y} \, , \qquad (A.3)$$

where $x_0, y_0 \to 0^+$. $h(x)$ is the step function, $h(x) = 1$ for $x > 0$ and $h(x) = 0$ for $x \le 0$. It has a transition function at $x = l_x$ and $y = l_y$:

$$\phi(l_x, y) = \phi(0, y) + 2\pi h(y - y_0), \quad \phi(x, l_y) = \phi(x, 0) + 2\pi h(x - x_0) \, . \qquad (A.4)$$

Note for an ordinary compact scalar the transition function is a constant multiple of $2\pi$. It can be interpreted as an ordinary compact scalar coupled to a background $B$ for the shift symmetry, given by

$$B = 2\pi h(y - y_0)\frac{dx}{l_x} + 2\pi h(x - x_0)\frac{dy}{l_y} \, , \qquad (A.5)$$

such that $\oint d\phi = \oint B_1$.

# B  Scalar or fermion on a leaf

## B.1  Scalar on a leaf

Consider complex scalar field $\phi^k$ coupled to a foliated $U(1)$ one-form gauge field $A_1^k$, with the kinetic term

$$S_{\text{kinetic}} = \int dt\,dx\,dy\,dz \sum_k |e^k(d-iA_1^k)\phi^k|^2 := \int \sum_k e^k(d+iA_1^k)\overline{\phi}^k \wedge \star\left(e^k(d-iA_1^k)\phi^k\right). \quad \text{(B.1)}$$

Note the kinetic term is invariant under the gauge transformation $A_1^k \to A_1^k + \alpha_1^k$ with $\alpha_1^k e^k = 0$. The gauge field $A_1^k$ thus couples to the current

$$\star j^k = ie^k \overline{\phi}^k \wedge \star(e^k d\phi^k) + \text{c.c.} . \quad \text{(B.2)}$$

The current satisfies $e^k \star j^k = 0$. For $e^k = dz$ on a flat spacetime, the current is

$$j_t^k = i\overline{\phi}^k \partial_t \phi^k + \text{c.c.}, \quad j_x^k = -i\overline{\phi}^k \partial_x \phi^k + \text{c.c}, \quad j_y^k = -i\overline{\phi}^k \partial_y \phi^k + \text{c.c}, \quad j_z^k = 0 . \quad \text{(B.3)}$$

The corresponding symmetry in the free scalar theory is the subsystem planar symmetry $\phi^k \to \phi^k e^{i\lambda(z)}$, or more generally $e^k d\lambda = 0$.

More generally, we can include a potential $V(\phi^k)$ for the scalar field. The potential that respects the symmetry takes the form $V(\phi^k) = f(\overline{\phi}^k \phi^k)$.

The discussion can be generalized to a real scalar $\varphi^k$, where $A_1^k$ is replaced by a foliated $\mathbb{Z}_2$ gauge field (which can be expressed as a $U(1)$ foliated gauge field with holonomy constrained to be 0 or $\pi$ mod $2\pi$).

**Discontinuity**  Since $A_1^k$ contributes the discontinuity $\delta(\mathcal{V}_k)^\perp$ in the kinetic term, for the kinetic energy to be finite, $e^k d\phi^k$ can at most have discontinuities. Thus the scalar field $\phi^k$ can have discontinuity $\delta(\mathcal{V}_k)^\perp$.

**Scalar of charge q > 1**  If the scalar field has charge $q$, we replace $A_1^k$ by $qA_1^k$. The $aU(1)$ foliated gauge theory with scalar of charge $q$ has $\mathbb{Z}_q$ electric symmetry that shifts the gauge field by a flat $\mathbb{Z}_q$ connection $A_1^k \to A_1^k + \lambda_1^k$, $\lambda_1^k = d\lambda_0^k/q$. We can couple the theory to background $\mathbb{Z}_q$ two-form gauge field $C_E^k$ using the electric symmetry. $C_E^k$ has the background gauge transformation

$$C_E^k \to C_E^k + d\lambda_1^k + \alpha_2^k, \quad A_1^k \to A_1^k + \lambda_1^k, \quad \alpha_2^k e^k = 0 . \quad \text{(B.4)}$$

We can also give the charge-$q$ scalar a potential $V(\phi^k)$. If the potential is chosen such that the scalar condenses, then the $U(1)$ foliated gauge field $A_1^k$ is Higgsed to $\mathbb{Z}_q$. Then the low energy theory is described by the foliated $\mathbb{Z}_q$ gauge theory

$$\frac{q}{2\pi} \sum_k dA_1^k B_2^k . \quad \text{(B.5)}$$

The Higgs phase still has the $\mathbb{Z}_q$ electric symmetry, with the background $C_E^k$ coupled as $\frac{q}{2\pi} \sum_k B_2^k C_E^k$.

## B.2 Fermion on a leaf

Consider a Dirac fermion $\psi^k$ coupled to foliated $U(1)$ one-form gauge field $A_1^k$, with the kinetic term

$$\int \sum_k i\overline{\psi}^k \gamma e^k \wedge \star(e^k(d - iA_1^k)\psi^k) , \tag{B.6}$$

where $\gamma = \gamma^\mu dx^\mu$. Note the kinetic term is invariant under $A_1^k \to A_1^k + \alpha_1^k$ with $\alpha_1^k e^k = 0$. The gauge field $A_1^k$ couples to the current

$$\star j^k = \overline{\psi}^k \star (\gamma e^k)e^k \psi^k . \tag{B.7}$$

The current satisfies $e^k \star j^k = 0$. For instance, if $e^k = dz$, the current is

$$j_\mu^k = \overline{\psi}^k \gamma_\mu \psi^k \text{ for } \mu = 0, 1, 2, \quad j_3^k = 0 . \tag{B.8}$$

The corresponding symmetry in the free fermion theory is the subsystem planar symmetry $\psi^k \to \psi^k e^{i\lambda(z)}$, or more generally $e^k d\lambda = 0$.

More generally, we can include a mass term and four-fermion interactions. If the theory has a real scalar $\varphi^k$, then we can include the Yukawa coupling

$$\sum_k m\overline{\psi}^k \psi^k + r(\overline{\psi}^k \gamma^\mu \psi^k)(\overline{\psi}^k \gamma_\mu \psi^k) + r'(\overline{\psi}^k \psi^k)^2 + g\varphi^k \overline{\psi}^k \psi^k . \tag{B.9}$$

**Discontinuity** Since $A_1^k$ can have discontinuity $\delta(\mathcal{V}_k)^\perp$, for the kinetic energy to be finite, $e^k d\psi^k$ can at most have discontinuity $\delta(\mathcal{V}_k)$. Thus $\psi^k$ can have discontinuity $\delta(\mathcal{V}_k)^\perp$.

## C Exactly solvable Hamiltonian model for $\mathbb{Z}_2$ electric model

In this appendix, we construct an exactly solvable Hamiltonian model for the $\mathbb{Z}_2$ electric model in (4.13). Consider a single foliation with $e^k = dx$, and $N = 2$. An exactly solvable model for the Hamiltonian can be constructed as follows. Integrating out $A^k, b$ in (4.13) gives $\mathbb{Z}_2$ gauge fields $B, a$ coupled as

$$\phi_4(a, B) = B \cup a \cup a , \tag{C.1}$$

where $B$ only has nonzero components $xy, xz$ but $B_{yz} = 0$, and we use $a \cup a = Sq^1 a$, which is $da/2$ for $da = 0$ mod 2. We have

$$\phi_4(d\phi, d\lambda) = d\phi_3(\phi, \lambda), \quad \phi_3(\phi, \lambda) = \lambda d\phi \cup d\phi . \tag{C.2}$$

An exactly solvable model for the SPT phase for the symmetry that shifts $\phi, \lambda$ is

$$H_{\text{SPT}} = -\sum_v X_v(-1)^{\int \phi_3(\phi+\tilde{v},\lambda)-\phi_3(\phi,\lambda)} - \sum_e X_e(-1)^{\int \phi_3(\phi,\lambda+\tilde{e})-\phi_3(\phi,\lambda)} , \tag{C.3}$$

where $\tilde{v}$ is the 0-cochain that equals 1 on the vertex $v$ and zero otherwise, and similarly $\tilde{e}$ is the one-cochain that equals 1 on edge $e$ and 0 otherwise. Explicitly,

$$H_{\text{SPT}} = -\sum_v X_v(-1)^{\int d\lambda \cup (\tilde{v} \cup d\phi + d\phi \cup \tilde{v} + \tilde{v} \cup d\tilde{v})} - \sum_e X_e(-1)^{\int \tilde{e} \cup d\phi \cup d\phi} . \tag{C.4}$$

Next, we gauge the shift symmetry of $\phi, \lambda$ to obtain the gauge theory for the gauge fields $a, B$. We introduce new qubits on edge and faces, acted on by Pauli matrices with a hat. We impose the Gauss law

$$X_v \prod \hat{X}_e = 1, \quad X_e \prod \hat{X}_f = 1 , \tag{C.5}$$

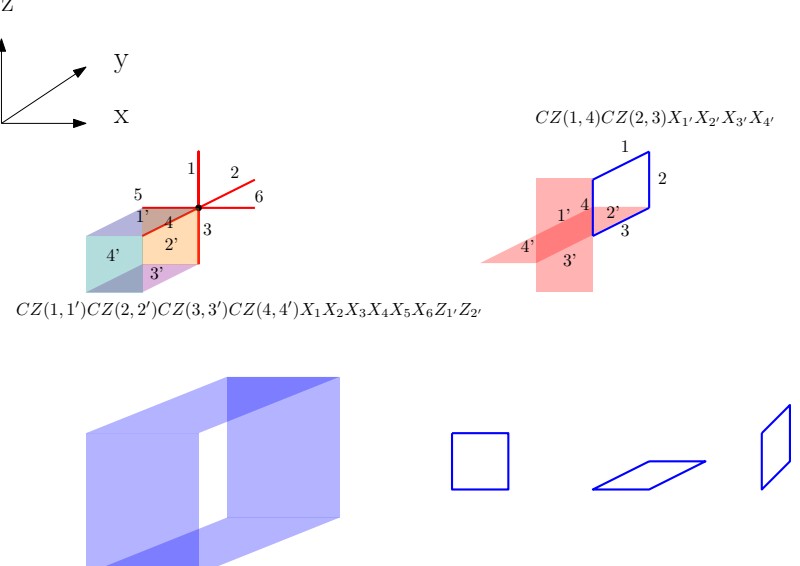

Figure 8: Hamiltonian model for the $\mathbb{Z}_2$ electric model. In the top row, the edges are labelled by numbers without a prime, while the faces are labelled by numbers with a prime. The operators in the top row only commute in the zero flux sector, which is the ground state subspace of the bottom-right plaquette operators.

where the product is over the adjacent edges and the adjacent faces. For the Hamiltonian to commute with the Gauss law, we replace $d\phi$ with $d\phi + a$, and $d\lambda$ with $d\lambda + B$. Then we can use the Gauss law to gauge fix $\phi, \lambda = 0$. We also include the flux term of the gauge fields.

The resulting Hamiltonian after gauging the symmetry is

$$H_{\text{gauged}} = -\sum_v \left(\prod \widehat{X}_e\right)(-1)^{\int B \cup (\widetilde{v} \cup a + a \cup \widetilde{v} + \widetilde{v} \cup d\widetilde{v})} - \sum_e \left(\prod \widehat{X}_f\right)(-1)^{\int \widetilde{e} \cup a \cup a}$$
$$- \sum_c \prod \widehat{Z}_f - \sum_f \prod \widehat{Z}_e \, . \tag{C.6}$$

The vertex, edge and the flux terms of the Hamiltonian are shown in Figure 8. The operators in the first line (top row in Figure 8) only commute in the zero flux sector, which is the ground state subspace of the bottom-right plaquette operators $\left(\sum_f \prod \widehat{Z}_e\right)$. Nevertheless, the ground state is exactly solvable and is the simultaneous ground state of all terms in the Hamiltonian. The Hamiltonian can be turned into a local commuting projector model by conjugating the first term (top-left in Figure 8) by ground state projectors of the last term (bottom-right in Figure 8).

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
