# Peer review of "Comments on Foliated Gauge Theories and Dualities in 3+1d"

_SciPost Physics, doi:SciPost Phys. 11, 032 (2021)_

## Round 2 · Referee Report · Anonymous (Referee 1) · 2021-7-3

Strengths

  1. The authors give a systematic field theory constructions of various known and new fracton models and SSPT. The global symmetries, anomalies, correlation functions, and discontinuities of the fields are analyzed carefully.

  2. The authors also construct the explicit lattice models for these field theories.

Report

The manuscript is beautifully and clearly written on an important subject. It also provides an elegant field theory description of a hybrid model recently constructed in [31]. I would highly recommend the publication of this manuscript.

Requested changes

  1. The function $h(z-z_0)$ is defined on p8 as a multiple of the step function, but I think (A.3) is only correct if $h(z-z_0)$ is literally the step function, i.e. $h(0^+)-h(0^-)=1$.

  • validity: top
  • significance: high
  • originality: high
  • clarity: high
  • formatting: excellent
  • grammar: excellent

Author:  Kevin Slagle  on 2021-07-15  [id 1569]

(in reply to Report 1 on 2021-07-03)

Thank you for the very positive report.

1) Thank you for pointing out this notation issue. We clarified it in the new draft.

---

## Round 2 · Referee Report · Anonymous (Referee 2) · 2021-7-8

Strengths

  1. The authors provide a solid analysis of foliated gauge theories, and enumerate the possible couplings to ordinary gauge theories. These theories are of great interest, and provides an elegant description of many fracton models constructed to date.

  2. The various constructions are very general, and encompass many previous examples in the literature, but also introduce new ones.

Weaknesses

  1. The paper could really benefit from adding more figures illustrating the excitations and mobilities in each model, and their corresponding fusion/braiding properties.

Report

This paper is a huge stepping stone towards the study of fracton phases. I would highly recommend the publication of this paper after the authors address some questions and take into consideration some minors suggestions listed below.

Requested changes

Below are some general questions I have for the paper

  1. Could the authors elaborate more about excitations (and their mobilities) that undergo 3-loop braiding in Sec. 2.6? A figure showing the shape of the excitations and an example braiding process (even for the 1-foliated case) would be very helpful.

  2. In Sec. 3.4, the authors introduce a fracton model that is not local unitary equivalent to X-cube (and seems to also argue that it is not foliated equivalent to X-cube either). Does the properties of this model match any known twisted X-cube models in the literature?

  3. Since the authors construct a model with non-abelian fractons, it would be nice to mention (and possibly compare/contrast to) some relevant works in the literature as well. There are a number of papers which have constructed fractons with $\mathbb{D}_8$ fusion rules.

  4. Could the authors please recheck the terms in the Hamiltonian in Fig. 7? The top two terms in the figure don't seem to commute (For example, when edge 1 of the left term coincides with edge 4 of the right term, when moving $X_1$ from the left term past $CZ(1,4)$ of the right term, it leaves $Z$ on edge 1 of the right term that is not cancelled by anything else.)

Here are some other minor suggestions that would improve the presentation of the paper

  1. It seems that the foliation index $k$ are sometimes implicitly summed over (for example in Lagrangians), while sometimes they are not e.g. $B^ke^k=0$. Making the sum over such index explicit would avoid possible confusion.

  2. In Sec. 2.2.1 and elsewhere, it would be nice to emphasize that the subscripts E and M refer to electric and magnetic, rather than the degree of the form, as used in the rest of the paper.

  3. The caption of Table 1 should probably say that the mobilities listed are for the specific case of $p_{ij}=1$ (unless the authors want to add more columns showing the mobilities for different possibilities of $p_{ij}$). Similarly in Figure 3, if some of $p_{ij}$ are zero, some of the excitations created will not be dipoles of a fracton.

  4. Since the translation between Lagrangian and lattice model contains expressions using cup products, it would be very helpful for the general reader if the authors could summarize the convention used on the lattice (or at the very least mention some useful references)

Other possible typos 1. In Eq. 2.4 and the paragraph above, should $B_1^k$ read $B_n^k$? 2. 't Hoot on p.10 should read 't Hooft 3. Some Z's in Figure 1 are probably missing daggers 4. In Eq. 5.23, should the $2$ be replaced by $N$?

  • validity: high
  • significance: high
  • originality: high
  • clarity: good
  • formatting: excellent
  • grammar: excellent

Author:  Kevin Slagle  on 2021-07-15  [id 1568]

(in reply to Report 2 on 2021-07-08)

Thank you for carefully reviewing our work and for the helpful suggestions for improvement.

general questions: 1) Thank you for pointing out this lack of clarity. We now describe these excitations in more detail.

2) That is an excellent question. This is currently an open question, which we hope will be revisited in the future.

3) Indeed, this is a useful connection to make. After analyzing the excitations, now conjecture that our $\mathbb{D}_8$ theory describes the non-abelian $D_4$ model in the ``Topological Defect Networks for Fractons of all Types'' work.

4) Thank you for pointing this out. Indeed, the top two rows only commute in the zero flux sector, which is the ground state subspace of the bottom-right plaquette operators. Nevertheless, the ground state is still exactly solvable and is the simultaneous ground state of all terms in the Hamiltonian. If desired, the Hamiltonian can be turned into a local commuting projector model by multiplying the top-left operators by ground state projectors of the bottom-right plaquette operators.

minor suggestions: 1) We appologize for the omission. We have added explicit foliation sums.

2) Thank you for the suggestion. We have added that clarification.

3) Thank you for pointing this out. We have clarified that Table 1 is only valid for $p_{kl} \not\equiv 0$ (mod $N$), and we have relabeled "fracton'' as $f^i$ excitations in Figure 3.

4) Indeed, that would be helpful. We added some useful references.

possible typos: Thank you for pointing out those typos. Regarding number 3, we have checked the Hamiltonian in Fig. 1 has GSD equal to 1 on a torus and that all of the operators commute. We do not think that any daggers are missing.

---

## Round 3 · Referee Report · Anonymous (Referee 2) · 2021-7-21

Report

I am happy with the changes and recommend the manuscript for publication. I only have some further minor comments for the authors to take into consideration below.

Requested changes

In Fig. 1, I agree that all the terms commute and that the GSD is equal to 1. However, could the authors clarify how the subsystem symmetries are defined on the lattice? If the symmetry is defined as an application of $X$ on rigid planes as usual, then the terms are not symmetric. Perhaps the authors are defining it as an alternating product of $X$ and $X^\dagger$ along the plane?

Regarding the description for three-loop braiding, the new paragraph added is still quite hard to follow and I would still encourage an accompanying figure. However, I will leave this to the discretion of the authors.

  • validity: high
  • significance: high
  • originality: high
  • clarity: good
  • formatting: excellent
  • grammar: excellent

Author:  Kevin Slagle  on 2021-07-27  [id 1619]

(in reply to Report 1 on 2021-07-21)

We have also created a new three-loop braiding figure (attached) for the next version.

Attachment:

Author:  Kevin Slagle  on 2021-07-21  [id 1600]

(in reply to Report 1 on 2021-07-21)

Thank you for clarifying. We have added daggers to Fig 1 so that the subsystem symmetry is simply a product of X operators along a plane. We also added a couple sentences to the figure caption to explicitly define the symmetry. Attached is the new figure.

Attachment:

BB_ZNSPT.pdf

---

## Round 3 · Author Response

We thank the referees and editors for their time and for thoroughly reviewing and improving our work.

---

## Round 3 · List of Changes

We have implemented many referee suggestions: 1. Clarify equation (A.3). 2. Elaborate on 3-loop braiding. 3. Conjecture D_8 model equivalence. 4. Clarify Fig. 7 commutivity. 5. Add explicit foliations sums. 6. Clarify E and M subscripts in Sec 2.2.1. 7. Clarify Table 1 caption. 8. Add cup product references. 9. Fixed typos

A more detailed overview of changes can be found in this diff: https://drive.google.com/file/d/1sH0v-6kFD2AIUPtQPCBovXYDNAUlAsC8

---

## Editorial Decision

published